# Genome-wide meta-analyses of restless legs syndrome yield insights into genetic architecture, disease biology and risk prediction

Restless legs syndrome (RLS) affects up to 10% of older adults. Their healthcare is impeded by delayed diagnosis and insufficient treatment. To advance disease prediction and find new entry points for therapy, we performed meta-analyses of genome-wide association studies in 116,647 individuals with RLS (cases) and 1,546,466 controls of European ancestry. The pooled analysis increased the number of risk loci eightfold to 164, including three on chromosome X. Sex-specific meta-analyses revealed largely overlapping genetic predispositions of the sexes ($r_g = 0.96$). Locus annotation prioritized druggable genes such as glutamate receptors 1 and 4, and Mendelian randomization indicated RLS as a causal risk factor for diabetes. Machine learning approaches combining genetic and nongenetic information performed best in risk prediction (area under the curve (AUC) = 0.82–0.91). In summary, we identified targets for drug development and repurposing, prioritized potential causal relationships between RLS and relevant comorbidities and risk factors for follow-up and provided evidence that nonlinear interactions are likely relevant to RLS risk prediction.

RLS is a prevalent, but underdiagnosed, chronic sensorimotor disorder, affecting up to 10% of the elderly population in Europe and North America[1,2]. Previous genome-wide association studies (GWAS) have identified 22 risk loci[3,4]. However, objective biomarkers for prediction or diagnosis are not available yet. Severely impairing sleep, RLS has a profound impact on daily functioning, overall health and quality of life. Long-term treatment options are scarce and require frequent adjustment due to side effects[2,5].

RLS is often comorbid with psychiatric disorders such as depression or anxiety as well as cardiovascular disorders, hypertension and metabolic conditions such as diabetes[2,6]. The extent to which these associations imply causal relations is unknown[7]. Epidemiological and clinical studies have consistently demonstrated the prevalence of RLS to be twice as high in women than in men[8,9]. The contribution of genetic factors to this difference has not been examined yet.

To address these shortcomings, we conducted a genome-wide association meta-analysis (GWAMA) of three independent GWAS. We integrated multiple layers of functional omics data to identify pathways and cell types relevant to RLS. Furthermore, our analyses included sex-stratified GWAS and a genetic investigation of the X chromosome. To facilitate translational research, we identified drug targets among candidate genes, used machine learning to enhance risk prediction and conducted extensive genetic correlation and Mendelian randomization (MR) analyses to identify risk factors.

## Results

### Pooled autosomal GWAS meta-analysis

We performed a meta-analysis of summary statistics from three GWAS for RLS, totaling 116,647 cases and 1,546,466 controls of European ancestry (Extended Data Fig. 1). The first GWAS (EU-RLS-GENE) was

✉e-mail: barbara.schormair@helmholtz-munich.de

conducted in affected individuals recruited by expert clinicians of the International EU-RLS-GENE consortium and ancestry-matched controls. The second GWAS (INTERVAL) was based on the INTERVAL study of blood donors in the United Kingdom, which used the Cambridge-Hopkins questionnaire to diagnose RLS. The third GWAS (23andMe) was conducted on the research participant base of 23andMe, identifying RLS by asking whether a diagnosis or treatment of RLS was received from a physician. Further details are provided in the Methods. Genetic correlations between the GWAS were strong but indicated some degree of heterogeneity, with pairwise genetic correlation ($r_g$) ranging between 0.70 and 0.76 (Extended Data Fig. 2), possibly due to differences in phenotyping of RLS as well as in source populations targeted for recruitment. Therefore, we used a multivariate GWAMA approach (Methods). After quality control, 9,196,648 variants with minor allele frequency (MAF) ≥ 1% were available for meta-analysis. We identified 161 RLS risk loci ($P < 5 \times 10^{-8}$) on the autosomes, confirming all known loci and adding 139 new loci (Extended Data Fig. 3a). Conditional analysis within each locus resulted in a total of 193 independent lead SNPs (Supplementary Table 1).

An LD score regression (LDSC) intercept of 1.072 (standard error (s.e.) = 0.013) with an inflation ratio of 0.064 (s.e. = 0.012) indicated that population stratification was negligible and that the inflation of the test statistics was driven by the polygenic architecture of RLS.

At the meta-analysis level, assuming a disease prevalence of 9%, the overall SNP-based heritability was estimated to be 0.20 (s.e. = 0.016) using LDSC (Methods). Because the meta-analysis included studies with different phenotyping methods, we also derived heritability estimates from the individual GWAS. LDSC-derived heritability in the most stringently phenotyped study, EU-RLS-GENE, was higher (0.26, s.e. = 0.038) than that in INTERVAL (0.17, s.e. = 0.051, $P_{EU\text{-}Interval} = 0.073$, two-sample two-sided $Z$-test) and 23andMe (0.14, s.e. = 0.011, $P_{EU\text{-}23andMe} = 0.0012$, two-sample two-sided $Z$-test). While the LDSC model showed the best fit, this trend was consistent with other estimation methods (Supplementary Table 2a).

### Sex-stratified autosomal GWAS and meta-analyses

To study sex-specific genetic effects, we conducted sex-stratified GWAS for the autosomes in each study and meta-analyzed the results (Extended Data Fig. 3b, representing 78,333 cases and 844,872 controls in women and 38,314 cases and 701,594 controls in men). Heritability was significantly higher for females in the meta-analysis ($h^2_{males} = 0.13$, s.e. = 0.012; $h^2_{females} = 0.32$, s.e. = 0.027; $P_{difference} = 1.9 \times 10^{-8}$, two-sided $Z$-test). The INTERVAL study was too small for reliable application of LDSC, but both other cohorts showed higher estimates for LDSC-derived heritability in females than in males ($P_{difference} = 0.07$ in EU-RLS-GENE; $P_{difference} = 0.09$ in 23andMe, two-sample two-sided $Z$-test; Supplementary Table 2b,c). Comparing the two sex-specific meta-analyses showed a high genetic correlation of 0.96 (s.e. = 0.018); however, the remaining small divergence was significant ($P = 0.044$, one-sample two-sided $Z$-test).

The sex-specific meta-analyses identified 58 independent lead SNPs in 50 risk loci in males and 155 SNPs in 130 loci in females (Supplementary Tables 3 and 4). Of these loci, 23 (two in males, 21 in females) were not genome-wide significant in the pooled analysis. To prioritize loci with robust sex differences, we tested the lead SNPs of the pooled meta-analysis for heterogeneity of effect sizes between males and females. This was statistically significant for six loci (Extended Data Table 1).

To understand the discrepancy between the heritability estimates of the two sexes despite their high genetic correlation, we ran a simulation study (Supplementary Note) modeling the impact of an environmental risk factor and of its interaction with the genetic predisposition to RLS ($G \times E$). The results obtained with the model including the $G \times E$ interaction recapitulated the situation observed in our real-world GWAS data very closely. This was the case for both binary and continuous environmental factors, with the binary risk factor showing a slightly better fit ($\log_{10}$ (Bayes factor) of 11.43 compared to 9.11). In line with this, the $G \times E$ model showed a closer fit ($P = 0.02$, two-sample two-sided $Z$-test) to the $h^2_{male}/h^2_{female}$ ratio observed in the pooled GWAS than the model without the $G \times E$ interaction (Extended Data Fig. 4). Furthermore, the impact of a $G \times E$ interaction on RLS was higher in females than in males with a $r_{G \times E(female)}/r_{G \times E(male)}$ ratio of 16.1 (95% CI = 7.09, 51.12).

### X-chromosomal meta-analyses

We performed pooled as well as sex-specific X chromosome-wide association study (XWAS) meta-analyses using EU-RLS-GENE and 23andMe data (Methods). Based on the pooled meta-analysis, SNP-based heritability $h^2_{pooled}$ carried by the X chromosome was 0.0035 (s.e. = 0.0010), with the sex-specific values again being lower in men ($h^2_{males} = 0.0032$, s.e. = 0.0018) than in women ($h^2_{females} = 0.0047$, s.e. = 0.0012; Extended Data Fig. 5 and Supplementary Table 5), but this difference was not significant ($P = 0.49$). Genetic correlation between the two sexes was high ($r_g = 0.926$, s.e. = 0.071, $P_{difference} = 0.29$, one-sample two-sided $Z$-test). Our analyses identified three independent risk loci for RLS on the X chromosome in the pooled data and one in the male-only data (Supplementary Tables 1 and 3).

### Replication of lead variants in additional datasets

We combined data from three additional cohorts to replicate the lead SNP associations of our meta-analyses (Methods): the discovery dataset of a previously published meta-analysis, a second research participant sample from 23andMe and a second set of blood donors from INTERVAL, totaling 29,028 cases and 398,815 controls. Despite the considerably smaller sample size, 71% of the lead SNPs from the pooled discovery meta-analysis were at least nominally significant in the replication dataset ($P < 0.05$) and there was a high positive correlation between the effect size estimates of the discovery stage and the replication dataset (Pearson's $r = 0.94$, $P < 2.2 \times 10^{-16}$; Extended Data Fig. 6a). The male- and female-specific analyses showed similar results (male, 67% of lead SNPs with $P < 0.05$, Pearson's $r = 0.97$, $P < 2.2 \times 10^{-16}$; female, 70%, Pearson's $r = 0.92$, $P < 2.2 \times 10^{-16}$; Extended Data Fig. 6b,c). A joint analysis of discovery and replication datasets revealed that all lead SNPs of the pooled, male-specific and female-specific meta-analyses reached Bonferroni-corrected significance (Supplementary Table 6).

### Functional annotation and biological interpretation

We performed gene set and cell type enrichment analyses based on the pooled meta-analysis (Methods). We used DEPICT to perform gene set enrichment analyses across the genome-wide significant risk loci and detected 319 gene sets with a false discovery rate (FDR) < 0.05 (Supplementary Table 7). These clustered in pathways, processes and structures related to neurodevelopment, neuron migration, axon guidance, synapse formation and signal transduction between neurons (Fig. 1a). An additional gene set enrichment analysis using MAGMA prioritized nine biological processes related to neuron migration and synapse formation with an FDR <0.05 (Supplementary Table 8). This supported the results from DEPICT and emphasizes the key role of neurodevelopmental processes in RLS biology (Fig. 1b).

We performed enrichment analyses to identify tissue and cell types involved in RLS. We first examined body-wide human gene expression data. The default analysis in DEPICT identified 24 of 209 tissue and cell types with significant enrichment (FDR < 0.05), 23 of which were central nervous system (CNS) tissues (Supplementary Table 9). Using GTEx version 8 as an independent validation dataset yielded highly comparable results (Supplementary Table 10). Therefore, we focused on higher-resolution single-cell sequencing datasets of the nervous system in mice, available for developmental and postnatal stages (Methods). Only neurons and neuroblasts showed statistically significant enrichment, while glial and endothelial cells,

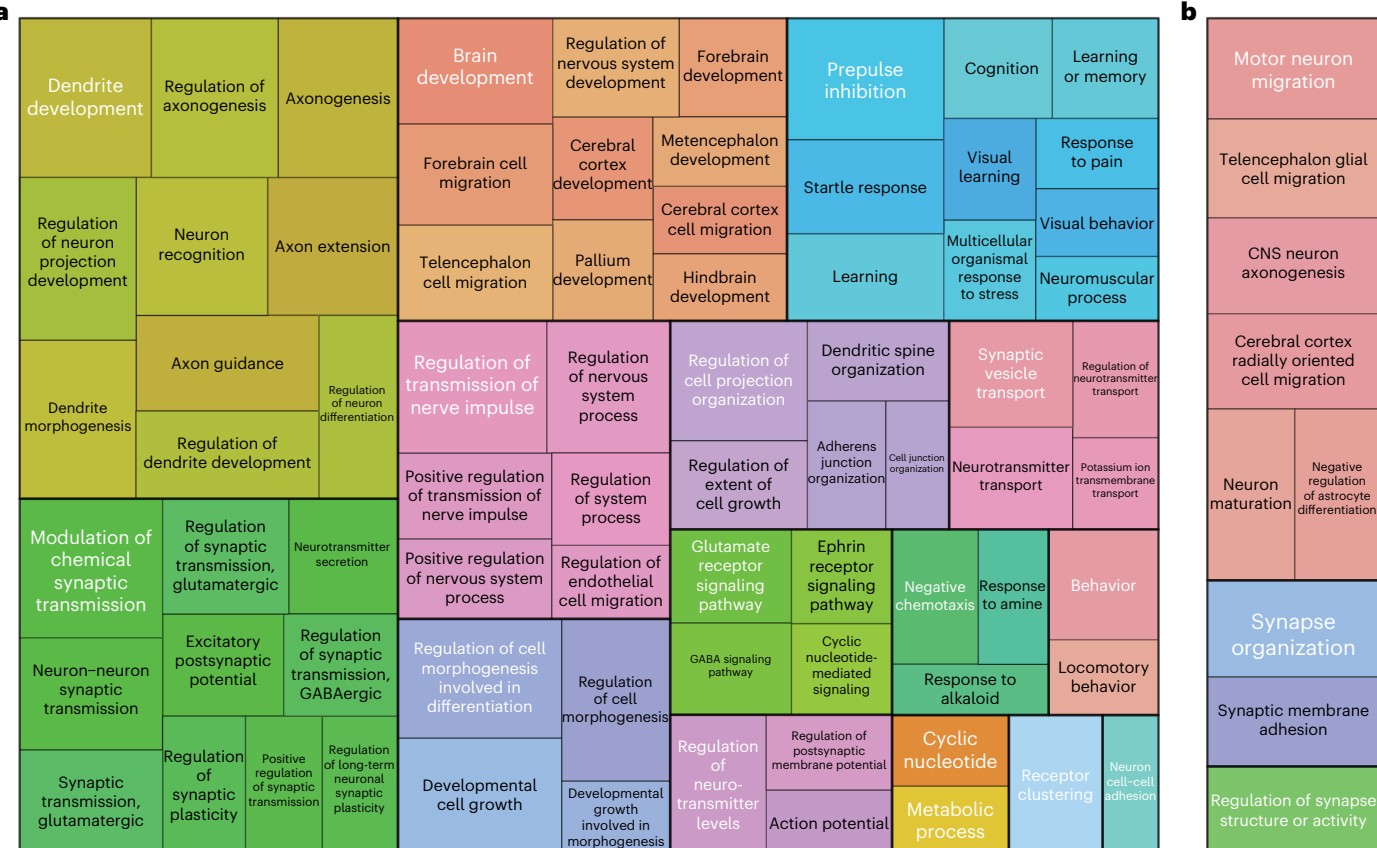

**Fig. 1 | Pathway enrichment analysis. a,b,** Treemaps of significantly enriched (FDR < 0.05, one-sample one-sided $Z$-test (DEPICT) or one-sided $t$-test (MAGMA)) pathways. Respective GO terms were clustered based on their semantic similarity (method: Wang, GOSemSim as implemented in the rrvgo package version 1.2.0) using results from DEPICT (**a**) and results from MAGMA (**b**). Terms are presented in rectangles. Coloring indicates the membership of a term in a specific cluster. In addition, each cluster is visualized by thick border lines. The size of each rectangle corresponds to the significance of the enrichment. The most significantly enriched term in each cluster was selected as the representative term and is displayed in white font.

for instance, did not (Fig. 2 and Supplementary Tables 11–14). We then dissected these cell types to identify specific anatomical regions and neurotransmitter classes (Fig. 2). We found cell types with statistically significant enrichment in all main compartments of the embryonic CNS: forebrain, midbrain, hindbrain and spinal cord. This was mirrored in the adult dataset, where cell types in the cerebrum, the cerebellum and the brainstem were highlighted. In most regions, both excitatory and inhibitory neuron types showed statistically significant enrichment, with glutamatergic neurons in the spinal cord showing the strongest enrichment. Overall, developmental-stage data yielded more robust enrichment than adult-stage data. Analyses in human datasets confirmed the enrichment in neuronal cell types and the higher level of significance obtained in the developmental datasets (Fig. 2 and Supplementary Tables 15 and 16). Again, excitatory and inhibitory neurons showed the highest enrichment. An additional analysis of bulk human brain transcriptome data from BrainSpan indicated an enrichment in the prenatal stage, but not the postnatal stage, underscoring a role for neurodevelopment in susceptibility to RLS (Supplementary Table 17).

We used diverse functional genomic annotation and fine-mapping approaches to build a sum score for ranking candidate causal genes within risk loci (maximum score = 12, Methods). Six loci contained no gene with a score above 2, 69 loci contained genes reaching a score of up to 6, and 89 loci contained genes with a score ≥ 7 (Supplementary Table 18). We focused further interpretation on the latter group. At 61 loci, there was a single independent lead SNP as well as a single top-scoring gene. These included six known loci,

strengthening previous reports (*MEIS1*, *PTPRD*, *SKOR1*, *NTNG1*, *CADM1* and *RANBP17*)[3,4,10–13]. Because drug repurposing is one of the fastest options for translating GWAS findings into patient care, we mapped the top-scoring genes against the druggable genome and identified 13 potential candidates targeted by existing compounds (Table 1). Among them, *GRIA1* and *GRIA4*, which encode subunits of AMPA-type glutamate ionotropic receptors, provided genetic evidence of a link between RLS and glutamate receptor function. Another interesting candidate is *CCKBR*, which encodes the predominant cholecystokinin receptor in the brain[14,15]. Our prioritization algorithm also listed *SLC40A1*, which had already been identified in the discovery stage of a previous study but had failed to replicate[4]. *SLC40A1* encodes ferroportin 1, the only known transporter for iron export from cells, being relevant for iron replacement therapies[16–18]. To evaluate whether iron-related traits and RLS shared causal variants in *SLC40A1*, we performed additional colocalization analyses using recently published GWAS of peripheral iron measures as well as quantitative susceptibility mapping (QSM) and T2* magnetic resonance imaging data as readouts for brain iron levels[19–21] (Supplementary Note). For the pallidum and the putamen, colocalization analysis pointed toward distinct causal variants (posterior probability for H3 hypothesis of coloc absolute Bayes factor analysis ($PP.H3.abf_{pallidum} \geq 96.1\%$ for QSM and $PP.H3.abf_{putamen} > 99\%$ for T2*), whereas results were inconclusive for the caudate nucleus. In other subcortical brain regions, the results were not statistically significant. For peripheral iron measurements, we saw a probability of >99% for different causal variants for both ferritin and total iron binding capacity and RLS. In general, our analyses suggest that the RLS

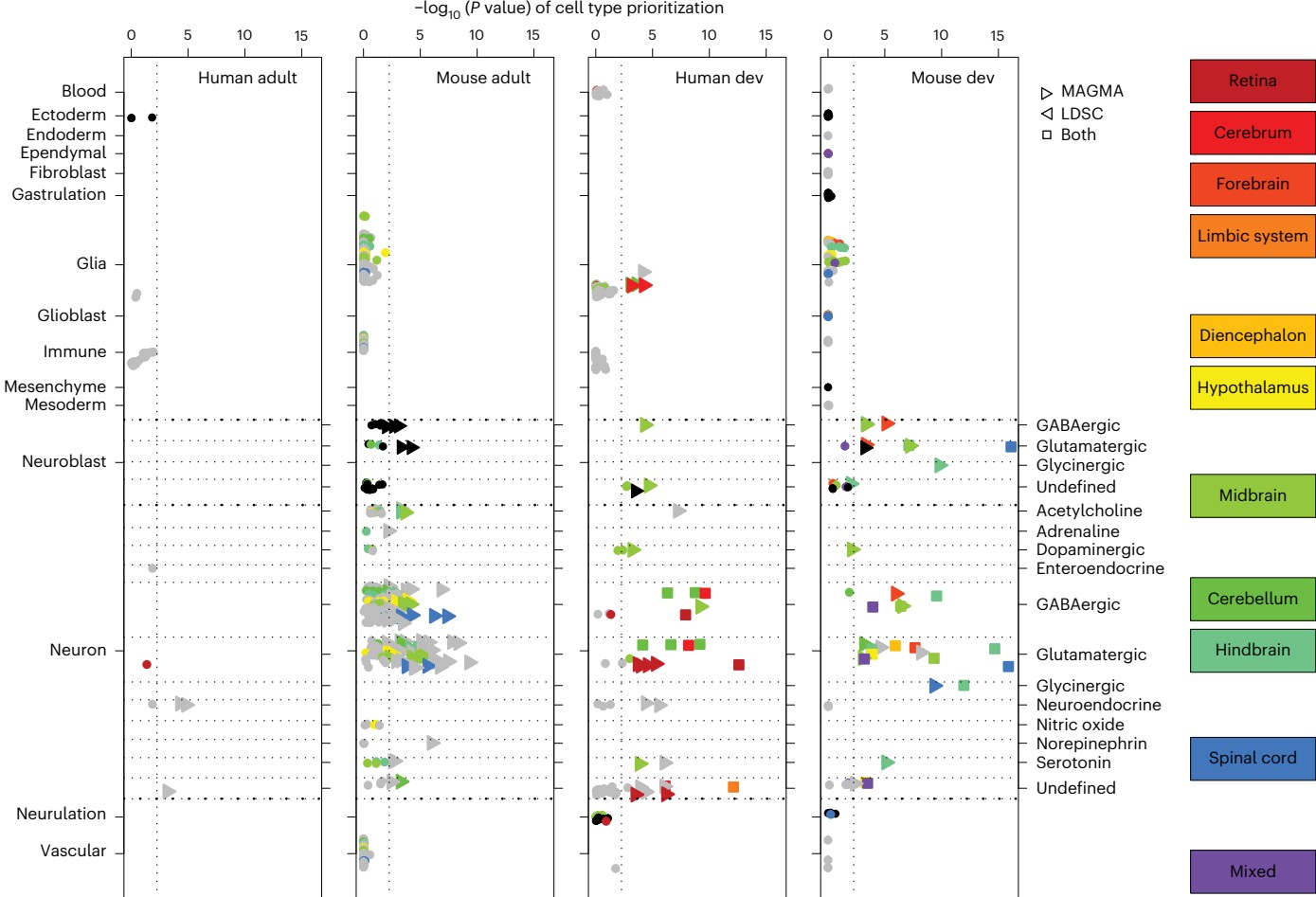

**Fig. 2 | Tissue and cell type enrichment analysis.** Cell type enrichment analysis results of mouse and human developmental (dev) and adolescent and adult CNS single-cell datasets. Cell types are annotated based on the class and subclass definitions used by the mouse brain atlases. We further annotated the neurons with their respective neurotransmitter type. Significance values (−log₁₀ (*P* value) and FDR, one-sample one-sided *Z*-test) are reported for MAGMA-based methods, using MAGMA_Celltyping for human adult data and CELLECT-MAGMA otherwise.

Gray and black dots indicate cell types not significant on the FDR level by both tools in any of the datasets; color alternates to separate neighboring cell types in the plot more clearly. The color code relates to the brain region where cells originated. Mixed refers to three cell types from the developmental-stage data, for which spatial mapping failed, resulting in an assignment to a mixture of brain regions (forebrain, midbrain, hindbrain) in varying proportions.

association in the *SLC40A1* locus is distinct from iron-related associations (Supplementary Table 19).

**Genetic correlation and MR analysis**

We performed a large-scale genetic correlation analysis followed by MR to discover potentially modifiable risk factors for RLS and to explore epidemiological or mechanistic overlaps with other diseases (Methods). Calculating genetic correlations with LDSC identified 1,054 of 2,649 analyzed traits and diseases as significantly correlated with RLS (FDR < 0.05; Supplementary Table 20). To factor in the complex interrelations between these traits, we performed bi-serial genetic correlation followed by weighted correlation network analysis of all 1,054 traits. This clustering yielded 11 modules, which reflected independent higher-level trait categories linked to RLS (Methods and Fig. 3a). The genetic correlation results strongly converged on RLS being associated with lower general physical as well as mental health. They confirmed epidemiological associations with increased body weight, depression, hypertension, cardiovascular disease, diabetes and sleep disturbances (Fig. 3b). However, they also provided evidence for less well-described associations of RLS with lower educational attainment, higher risk of asthma and diseases of the digestive system. In line with the increased prevalence in females, we identified a cluster of female-specific traits

such as age of first childbirth, hysterectomy, oophorectomy and excessive menstruation (blue module, Fig. 3a,b and Supplementary Table 20).

We performed MR to infer potential causal relationships between RLS and representative traits from these clusters (Fig. 4 and Supplementary Table 21). RLS as a common and complex disease is characterized by phenotypic heterogeneity and likely entails genetic pleiotropy, necessitating cautious interpretation of MR results. Therefore, we used the latent heritable confounder MR (LHC-MR) approach for the primary analysis, which is a robust method designed to account for pleiotropy and potential confounding (Methods). We confirmed known unidirectional and bidirectional relations, for example, that the number of live births significantly increased the risk of RLS or that insomnia symptoms and RLS were bidirectionally linked[8,9,22,23].

For other traits, LHC-MR indicated relationships being causal rather than due to confounding. In terms of unidirectional relationships, RLS showed a significant effect (defined as $P_{FDR} < 0.05$) on type 2 diabetes with an effect estimate of $a_{RLS \rightarrow diabetes2} = 0.99$ (s.e. = 0.06, $P_{FDR} = 1.5 \times 10^{-68}$) and significant likelihood-ratio tests (LRTs) for effects being only causal ($P_{LRT\_causal\_only} = 8.5 \times 10^{-28}$) and effects only of RLS on type 2 diabetes ($P_{LRT\_only\_RLS \rightarrow diabetes2} = 2.9 \times 10^{-40}$). Unidirectional causal links to RLS with strong evidence were fresh fruit intake (decreased RLS risk with $a_{fruit \rightarrow RLS} = -0.33 \pm 0.08$, $P_{FDR} = 0.0002$, $P_{LRT\_causal\_only} = 2.2 \times 10^{-5}$,

## Table 1 | Drug repurposing options for top-scoring genes

| GWAS locus lead SNP | | | Prioritized gene (score) | DrugBank-listed drugs or compounds | Druggability tier |
|---|---|---|---|---|---|
| ID | Position | P value | | | |
| rs10895816 | 11:105,285,122 | $1.16×10^{-25}$ | GRIA4 (10) | Talampanel, glutamic acid, CX-717 | 1 |
| rs10839553 | 11:6,350,791 | $7.65×10^{-16}$ | CCKBR (7) | Pentagastrin, cholecystokinin | |
| rs10038916 | 5:153,098,094 | $9.99×10^{-16}$ | GRIA1 (9) | Perampanel, lamotrigine, talampanel, glutamic acid, CX-717, CX516, tianeptine | |
| rs306960 | 8:142,005,245 | $1.18×10^{-13}$ | PTK2 (9) | Fostamatinib, endostatin | |
| rs12693542 | 2:190,445,848 | $1.35×10^{-13}$ | SLC40A1 (9) | Ferrous sulfate, tetraferric tricitrate decahydrate | |
| rs824920 | 2:222,786,280 | $1.35×10^{-12}$ | EPHA4 (7) | Fostamatinib | |
| rs714522 | 23:24,686,539 | $2.26×10^{-8}$ | POLA1 (7) | Fludarabine, clofarabine, cladribine, nelarabine | |
| rs2067133 | 5:102,364,542 | $7.59×10^{-19}$ | PAM (11) | Copper, vitamin C | 2 |
| rs17123518 | 20:31,248,265 | $1.88×10^{-11}$ | DNMT3B (7) | Decitabine | |
| rs56350804 | 2:217,560 | $3.55×10^{-11}$ | ACP1 (10) | Adenine | |
| rs72718216 | 14:69,455,773 | $3.23×10^{-34}$ | ACTN1 (9) | Copper, human calcitonin | 3 |
| rs11142701 | 9:73,762,953 | $6.01×10^{-15}$ | TRPM3 (11) | Primidone | |
| rs326779 | 11:29,617,859 | $3.81×10^{-11}$ | KCNA4 (7) | Dalfampridine | |

Genes are named according to Ensembl gene name nomenclature and are mapped to the druggability tiers as provided by Finan et al.[33]. For each gene, the corresponding risk locus is indicated with its respective lead SNP and the corresponding two-sided P value from the pooled N-weighted genome-wide association meta-analysis (N-GWAMA). Approved and investigational drugs and small compounds targeting the products of these genes were extracted from the DrugBank Online database (release 5.1.8, https://go.drugbank.com/). ID, dbSNP rsID; position, chromosome:position on GRCh37 (hg19); score, sum score of gene prioritization in risk loci.

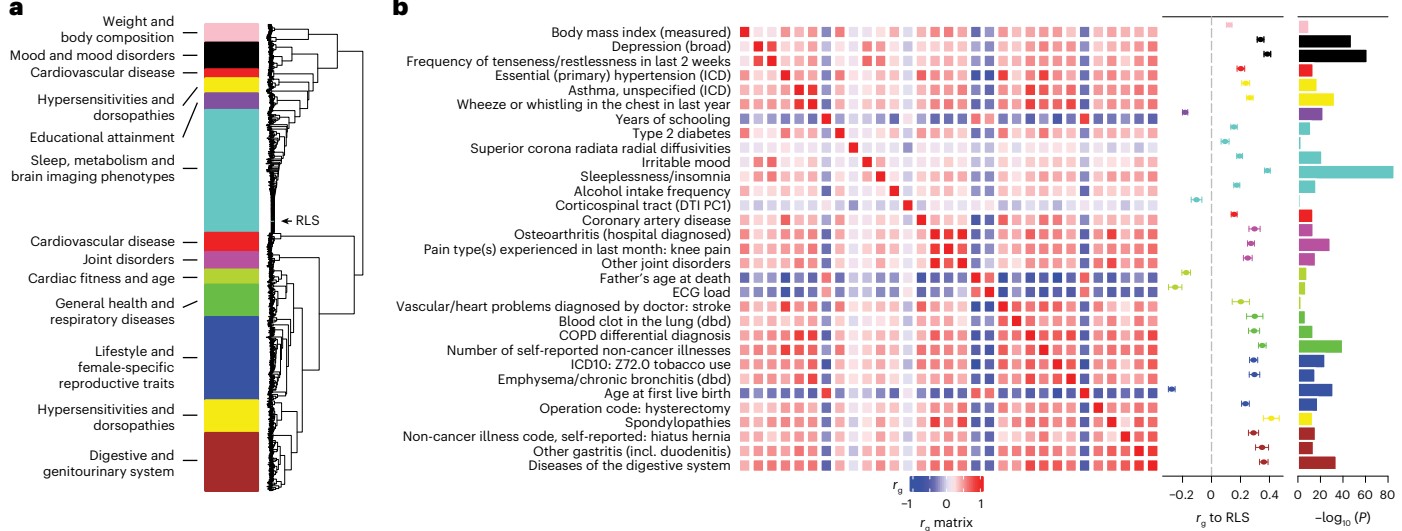

**Fig. 3 | Genetic correlation analysis. a**, Hierarchical clustering by a weighted correlation matrix analysis with the WGCNA package identified 11 modules. These were named with umbrella terms reflecting the type of traits contained in the respective module. **b**, Between-trait correlation matrix and genetic correlation with RLS for traits reflecting submodules identified in the 11 modules by consensus manual inspection of clusters. These traits were taken forward to MR analysis. The correlation matrix ($r_g$ matrix) indicates genetic correlation between the individual traits. Genetic correlation of each trait with RLS is indicated in the second column ($r_g$ to RLS); the significance of this correlation is reported in the third column ($-\log_{10}$ (P value), one-sample two-sided Z-test). COPD, chronic obstructive pulmonary disease; DTI PC1, diffusion tensor imaging principal component 1; ECG, electrocardiogram; dbd, diagnosed by doctor; ICD, ICD-10 coded hospital inpatient diagnosis; incl., including.

$P_{\text{LRT\_only\_fruit}\to\text{RLS}} = 2.3 \times 10^{-5}$) and being tense or highly strung as well as having had a headache in the last month (elevated RLS risk with $a_{\text{tense}\to\text{RLS}} = 0.44 \pm 0.06$, $P_{\text{FDR}} = 8 \times 10^{-12}$, $P_{\text{LRT\_causal\_only}} = 8.6 \times 10^{-9}$, $P_{\text{LRT\_only\_tense}\to\text{RLS}} = 4.2 \times 10^{-8}$ and $a_{\text{headache}\to\text{RLS}} = 0.37 \pm 0.08$, $P_{\text{FDR}} = 2.9 \times 10^{-5}$, $P_{\text{LRT\_causal\_only}} = 1.2 \times 10^{-8}$, $P_{\text{LRT\_only\_headache}\to\text{RLS}} = 6.9 \times 10^{-7}$). Significant bidirectional relations with evidence of only causal effects were found for five traits (all with $P_{\text{LRT\_causal\_only}} < 0.05$; Fig. 4 and Supplementary Table 21): ease of getting up in the morning lowered RLS risk ($a_{\text{ease}\to\text{RLS}} = -0.3 \pm 0.06$) and vice versa ($a_{\text{RLS}\to\text{ease}} = -0.09 \pm 0.02$). The frequency of tenseness or restlessness in the last 2 weeks as

well as two traits reflecting lung function increased RLS risk and vice versa, with a stronger effect on RLS ($a_{\text{tenseness}\to\text{RLS}} = 0.62 \pm 0.07$, $a_{\text{RLS}\to\text{tenseness}} = 0.11 \pm 0.02$, $a_{\text{COPD-differential-diagnosis}\to\text{RLS}} = 0.38 \pm 0.06$, $a_{\text{RLS}\to\text{COPD-differential-diagnosis}} = 0.12 \pm 0.03$), while, for self-reported osteoarthritis, RLS had the stronger effect ($a_{\text{osteoarthritis}\to\text{RLS}} = 0.46 \pm 0.19$, $a_{\text{RLS}\to\text{osteoarthritis}} = 0.18 \pm 0.04$). We performed inverse-variance weighted (IVW)-MR analyses with Steiger filtering and MR-Egger intercept assessment as a secondary analysis. The results were consistent for 14 traits, which included the unidirectional link between RLS and type 2 diabetes (Fig. 4 and Supplementary Table 22).

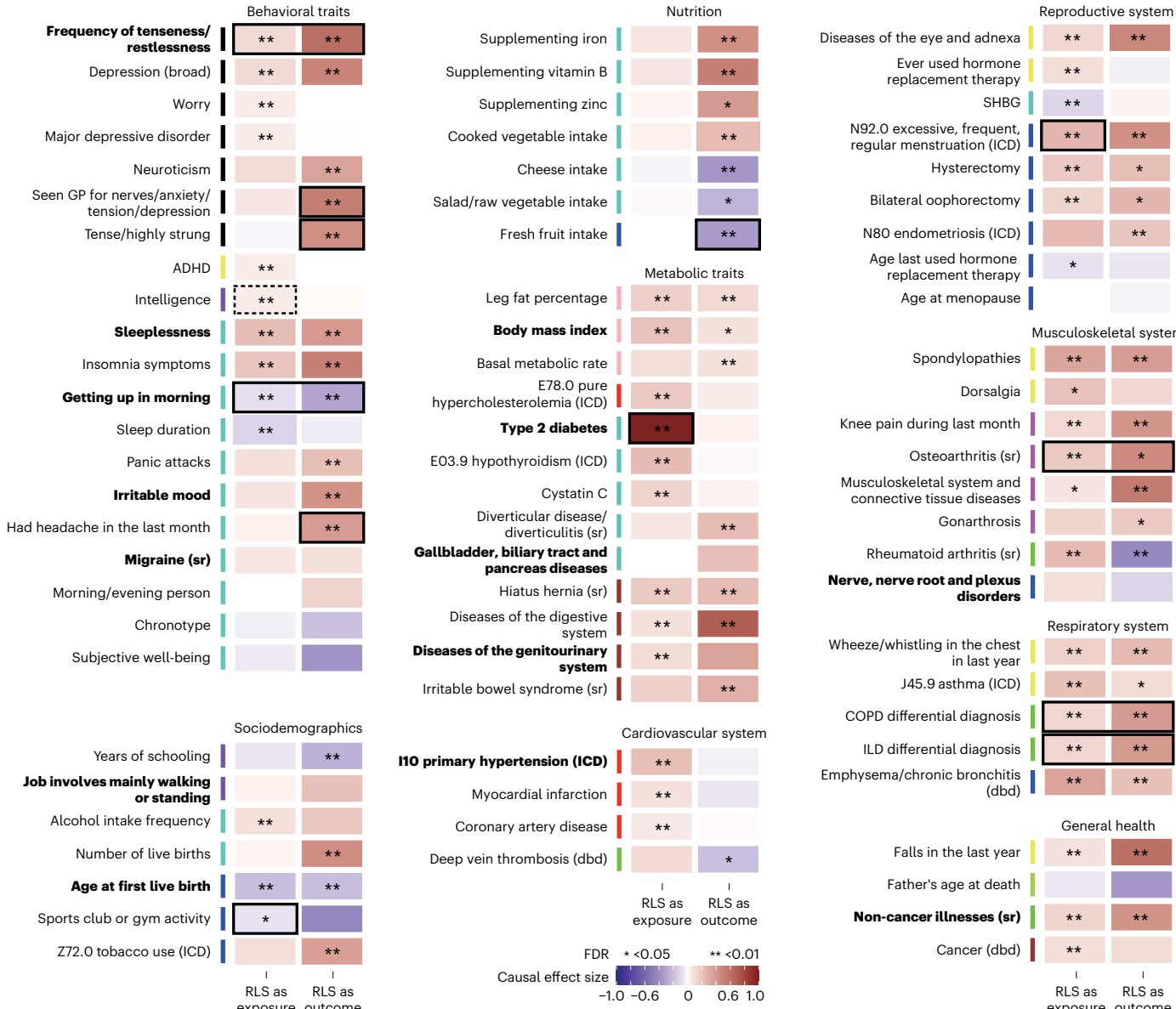

**Fig. 4 | MR analysis.** LHC-MR results for bidirectional MR between RLS and selected traits. The causal effect size is color coded, with dark blue indicating strong negative effects and dark red indicating strong positive effects. Significance of LHC-MR is reported as the FDR (LRT). Black borders mark traits for which LHC-MR analysis provided evidence for causal-only effects contributing to the relationship between the traits ($P_{LRT\_causal\_only} < 0.05$). Dashed black borders mark traits with evidence for confounding effects only in LHC-MR ($P_{LRT\_latent\_only} < 0.05$). Bold text indicates traits that showed consistent results in the IVW-MR analysis (one-sample two-sided $Z$-test, $P_{FDR\_filter} < 0.05$ for significant effects and <0.05 for nonsignificant effects; Supplementary Table 22). ADHD, attention-deficit–hyperactivity disorder; GP, general practitioner; ILD, interstitial lung disease; SHBG, sex hormone binding globulin; sr, self-reported; $P_{LRT}$, $P$ value of the LRT from LHC-MR.

Considering the proposed involvement of brain iron homeostasis in RLS[24] and *SLC4OA1* as a candidate gene in our GWAMA, we also investigated peripheral and brain iron traits. Both genetic correlation and MR analyses did not reveal strong effects (Supplementary Tables 21 and 23). Only white matter hyperintensity measured by T2* was significantly correlated with RLS in the full dataset ($r_g = 0.126$, s.e. = 0.046, $P = 0.0065$, $P_{FDR} = 0.016$, one-sample two-sided $Z$-test). LHC-MR revealed a significant effect of peripheral calculated transferrin levels on RLS; however, this appears to be largely attributable to confounding factors ($P_{LRT\_latent\_only} = 0.005$).

**Development and validation of a risk prediction model**

We assessed the predictive performance of basic linear models as well as that of models integrating interaction effects and time-dependent effects using genetic data and basic demographic variables such as age, sex and age of disease onset (Methods and Supplementary Note). We employed three classes of models, generalized linear models (GLMs) with or without interaction terms, random forest (RF) models and deep neural network (DNN) models, implemented as a binary or a time-to-event (survival) classifier. Genetic risk was calculated as a polygenic risk score (PRS) using individual dosages of 216 genome-wide significant SNPs (PRS.lead), because this score showed better performance than a score using genome-wide data (LDpred2) with an area under the receiver operator characteristic curve (AUC) of $AUC_{LDpred2} = 0.66 \pm 0.019$ compared to $AUC_{PRS.lead} = 0.73 \pm 0.018$ ($P = 0.0056$, two-sample two-sided $Z$-test).

Overall, the machine learning survival classifier models considering nonlinear interactions and time-varying effects performed

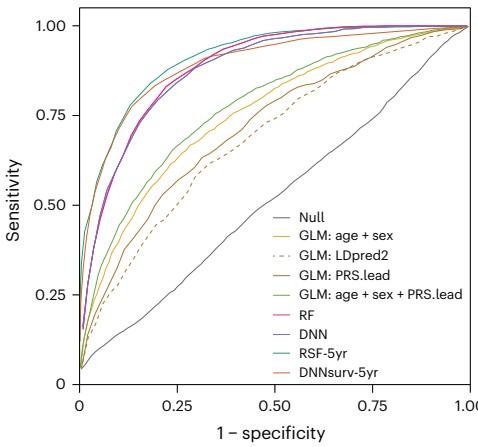

**Fig. 5 | Risk prediction.** Receiver operator characteristic curve showing the performance of different models used to predict RLS risk in the synthetic population representing the EU-RLS-GENE cohort. Null refers to the model including only the intercept ($y \approx 1$). GLM:age + sex refers to the model including age, sex and principal components (PCs). GLM:LDpred2 refers to the model including the genome-wide PRS calculated with LDpred2-auto. GLM:PRS. lead refers to the model including the PRS based on 216 lead SNPs. GLM:PRS. lead + age + sex refers to the model including age, sex and the PRS based on 216 lead SNPs. RF refers to the RF model. DNN refers to the DNN model. RSF-5yr refers to the RF survival analysis for a 5-year period. DNNsurv-5yr refers to the DNN survival analysis for a 5-year period.

best (Fig. 5). The random survival forest model (RSF-5yr; 5-year period) and the DNN survival model (DNNsurv-5yr) showed comparable performance: $AUC_{RSF-5yr} = 0.91 \pm 0.008$ compared to $AUC_{DNNsurv-5yr} = 0.90 \pm 0.012$ in the EU-RLS-GENE dataset and $AUC_{RSF-5yr} = 0.87 \pm 0.005$ compared to $AUC_{DNNsurv-5yr} = 0.86 \pm 0.012$ in the INTERVAL dataset. Additional performance metrics such as odds ratio (OR) and area under the precision–recall curve yielded the same trends (Supplementary Table 24).

We also evaluated the contribution of the interaction effects to the model performance either directly (GLMs) or indirectly by calculating the incremental gain in explained variance for the DNN and RF models (Nagelkerke's pseudo-$R^2$; Methods). For the GLM, we found a significant interaction between PRS and age ($\beta = -0.47$, s.e. = 0.08, $P = 4.3 \times 10^{-9}$, one-sample two-sided $Z$-test). The impact of the PRS was significantly lower in the 60+ age group ($OR_{overall} = 5.05$ (4.69–5.45), $OR_{60+} = 3.70$ (3.27–4.19), $P_{difference} = 2.6 \times 10^{-5}$, two-sample two-sided $Z$-test). We did not find a significant sex difference in overall PRS effects ($OR_{male} = 4.70$ (4.16–5.31), $OR_{female} = 5.28$ (4.80–5.80), $P_{difference} = 0.141$, two-sample two-sided $Z$-test), even though the effect of sex was highly significant (OR = 2.54 (2.33–2.78), $P = 1.93 \times 10^{-94}$, one-sample two-sided $Z$-test). In the best-performing RF and DNN binary classification models, pseudo-$R^2$ was 0.329 (s.e. = 0.003) and 0.324 (s.e. = 0.005), almost 1.5 times higher than in the GLM ($R^2 = 0.221$, s.e. = 0.003). The time-to-event classifier models showed a further increase in $R^2$ by approximately 10% for both models ($R^2_{RSF-5yr} = 0.363$, s.e. = 0.004; $R^2_{DNNsurv-5yr} = 0.354$, s.e. = 0.005). Overall, nonlinear relationships and interactions accounted for 39.1% (s.e. = 1.96%) of the explained variance.

## Discussion

Performing the largest meta-analysis of RLS GWAS to date, we have increased the number of known risk loci eightfold. We included three cohorts, representative of commonly used strategies to assess behavioral phenotypes, ranging from in-person interviews to a single online question. They also reflect the breadth of target populations for recruitment into GWAS, including clinical cohorts as well as samples from the general population. Despite this heterogeneity, genetic correlations were strong between the cohorts, justifying their combination in a multivariate meta-analysis.

We investigated sex-specific genetic susceptibility in RLS. While the heritability was significantly higher in women, the genetic correlation between the sexes was close to one. Results from our simulation study pointed to an unobserved environmental risk factor and corresponding gene–environment interactions driving the difference in heritability. Our analyses emphasize the importance of tracking environmental exposures in genetically susceptible individuals and may motivate re-interpretation of previous observations in RLS, for example, of parity potentially driving the higher prevalence observed in women[8,9]. In line with the high genetic correlation between the sexes, there were only six loci where risk variants showed significant sex differences in effect size. An additional two loci in males and 21 loci in females were genome-wide significant in only one sex but did not reach significance in the between-sex heterogeneity tests. With larger sample sizes, some of these may turn out to be true sex-specific association signals.

Our enrichment analyses corroborate results from earlier GWAS of RLS by prioritizing CNS tissues and primarily pathways linked to neurodevelopment and neurotransmission[3]. Interestingly, the enrichment effects were consistently stronger in fetal and prenatal datasets. This suggests that development may represent a critical period in which genetic contributors to RLS susceptibility act on the activity, connectivity or composition of neurons in the CNS. Analyses in developmental mouse CNS single-cell data prioritized excitatory glutamatergic neurons in the spinal cord, hindbrain, midbrain and forebrain but also γ-aminobutyric acid (GABA)ergic neurons in at least the midbrain and hindbrain. This diversity was reflected in the adult dataset, with again mostly excitatory neurons showing enrichment. Overall, the diversity of cell types and structures with significant enrichment corresponds to the complex phenotype of RLS, which includes sensory and motor symptoms as well as a circadian pattern. Unfortunately, the current scarcity of high-resolution data limits the ability of our study to validate these observations in humans. Tissue enrichment analysis depends on the methodology as well as on the composition of the datasets. Specifically, definite exclusion of cell types is difficult as they may not have been represented in the dataset. We tried to address these limitations by using two different enrichment methods as well as several datasets.

Interestingly, except for the prioritization of *SLC40A1* (ferroportin), we did not identify strong links between iron metabolism and genetic risk factors for RLS in our pathway and genetic correlation analyses. However, the T2* and QSM values we used as surrogates for brain iron content are differentially influenced by iron and myelin; therefore, future magnetic resonance imaging GWAS with higher anatomical resolution may allow better dissection of genetic effects involved in iron and myelin content[25,26]. Moreover, we cannot rule out an incomplete representation of brain or general iron homeostasis in the currently available pathway definitions.

Our study provides discoveries relevant for advancing clinical care in RLS. We identified several genes that are druggable and in some cases targets of known drugs. For example, the prioritization of two glutamate receptors suggests that the efficacy of anticonvulsants in RLS should be re-assessed. Small open trials have shown good response to glutamate receptor antagonists such as perampanel or lamotrigine in RLS[27,28]. The benefit of α2δ ligands such as pregabalin or gabapentin adds further evidence that anti-epileptic drugs could be an additional therapeutic option[29]. Investigation into a completely new line of treatment is suggested by the prioritization of the cholecystokinin B receptor, a neuropeptide receptor that has been linked to pain modulation and anxiety-related behavior[15,30]. Furthermore, our genetic correlation and MR analyses identified relationships of potential medical relevance between RLS and several traits. In line with previous reports, the strongest genetic correlations with RLS were observed for insomnia symptoms and for depression[22,23]. MR analysis showed bidirectional

effects, with the full model (causal as well as confounding effects) performing best. Probably, both pleiotropic genetic effects as well as the presence of RLS cases in the depression and insomnia cases and vice versa are involved. Disentangling the contributions of shared genetics and of case misclassification to this relationship will require large datasets with high-quality phenotyping of both insomnia and RLS. We saw a robust and significant unidirectional relationship of RLS with type 2 diabetes, with consistent results between LHC-MR and standard IVW-MR. The causal-only-effect model performed best in LHC-MR, suggesting that this link from RLS to diabetes is unlikely due to a heritable confounder. Thus far, cross-sectional and clinical studies have yielded inconsistent results regarding the causal relationship between RLS and type 2 diabetes[31]. Our MR analyses support a causal effect of RLS increasing the risk of type 2 diabetes. We found likely causal, albeit bidirectional relationships between RLS and osteoarthritis and between RLS and diseases of the respiratory system. Clinical or epidemiological studies on RLS in these disorders are limited or even non-existent at present; therefore, patients could benefit from increased awareness and research activities. The beneficial effect of modifiable behaviors on reducing the risk of RLS is underscored by findings that a healthy lifestyle, for example, fresh fruit consumption, is linked to lower RLS risk. Due to the inherent limitations of MR analysis, these results should not be overinterpreted. Even though the LHC-MR approach seems robust across a range of scenarios with different violations of the MR assumptions, it has its own drawbacks[32]. Therefore, we advise leveraging our findings to inform future clinical and epidemiological research aimed at gathering further evidence to support causality.

Predicting the likelihood of developing RLS is crucial for targeted disease-prevention strategies. We compared traditional PRSs to more advanced machine learning approaches integrating interaction and nonlinear effects. The latter showed superior performance compared to simple PRS-only or PRS-plus-linear interactions models. In our simulation study with only limited phenotypic data, the RF and DNN approaches provided comparable results. Enhanced phenotypic data may amplify the effectiveness of DNNs for predictive purposes. Two aspects limited our options for risk prediction. First, the definitive RLS cases (diagnosed by face-to-face interviews) with individual-level data required for developing the models had no detailed clinical data. Second, they were part of a case–control cohort and therefore do not reflect the general population structure, which necessitated creating a simulated dataset from the original data. Nevertheless, we were able to achieve an AUC of up to 91% for the 5-year prediction window with the machine learning approaches and validated our results in the INTERVAL study, where the performance was comparable with an AUC of up to 87%.

Collectively, our study marks a substantial advance in deciphering the genetic basis of RLS and paves the way for improving treatment and prevention strategies. We acknowledge two important limitations. First, biobank-scale longitudinal datasets with detailed medical and lifestyle information and high-quality RLS phenotyping are lacking. This type of data is needed to dissect the relationships discovered by genetic correlation and MR analyses as well as to study the roles of age, sex and other environmental effects and their interactions in shaping the risk and course of disease. Second, large-scale GWAS for RLS are currently limited to populations of European ancestry. An extension to non-European populations is imperative to improve genetic fine-mapping at shared loci and to adapt disease concepts to these populations with respect to non-shared genetics.

## Online content

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

Barbara Schormair [1,2,78] ✉, Chen Zhao[1,2,78], Steven Bell [3,4,5,78], Maria Didriksen [6,7], Muhammad S. Nawaz[8], Nathalie Schandra[1,2], Ambra Stefani [9], Birgit Högl[9], Yves Dauvilliers[10], Cornelius G. Bachmann[11,12], David Kemlink[13], Karel Sonka[13], Walter Paulus[14], Claudia Trenkwalder[15,16], Wolfgang H. Oertel[1,17], Magdolna Hornyak[18], Maris Teder-Laving[19], Andres Metspalu[19], Georgios M. Hadjigeorgiou [20], Olli Polo[21], Ingo Fietze[22], Owen A. Ross [23], Zbigniew K. Wszolek [24], Abubaker Ibrahim[9], Melanie Bergmann[9], Volker Kittke [1,2], Philip Harrer[1,2], Joseph Dowsett[6], Sofiene Chenini[10], Sisse Rye Ostrowski [6,25], Erik Sørensen[6], Christian Erikstrup [26,27], Ole B. Pedersen [25,28], Mie Topholm Bruun [29], Kaspar R. Nielsen[30], Adam S. Butterworth [31,32,33,34,35], Nicole Soranzo [33,36,37], Willem H. Ouwehand [36,38,39], David J. Roberts[33,40,41], John Danesh[31,32,33,34,35,37], Brendan Burchell[42], Nicholas A. Furlotte[43], Priyanka Nandakumar[43], 23andMe Research Team*, D.E.S.I.R. study group, Christopher J. Earley[44], William G. Ondo[45], Lan Xiong[46,47], Alex Desautels[48,49], Markus Perola[50,51], Pavel Vodicka[52,53,54], Christian Dina [55], Monika Stoll [56], Andre Franke [57], Wolfgang Lieb [58], Alexandre F. R. Stewart [59], Svati H. Shah[60,61], Christian Gieger [62,63], Annette Peters [62,64,65], David B. Rye[66], Guy A. Rouleau[46,47,67], Klaus Berger[68], Hreinn Stefansson[8], Henrik Ullum[69], Kari Stefansson[8], David A. Hinds [43], Emanuele Di Angelantonio[31,32,33,34,35,70,79], Konrad Oexle [1,2,71,79] & Juliane Winkelmann [1,2,72,73,79]

[1]Institute of Neurogenomics, Helmholtz Zentrum München, German Research Center for Environmental Health, Neuherberg, Germany. [2]Institute of Human Genetics, TUM School of Medicine and Health, Technical University of Munich, Munich, Germany. [3]Department of Oncology, University of Cambridge, Cambridge, UK. [4]Department of Clinical Neurosciences, University of Cambridge, Cambridge, UK. [5]Cancer Research UK Cambridge Institute, Li Ka Shing Centre, University of Cambridge, Cambridge, UK. [6]Department of Clinical Immunology, Copenhagen University Hospital, Rigshospitalet, Copenhagen, Denmark. [7]Department of Neuroscience, University of Copenhagen, Copenhagen, Denmark. [8]deCODE Genetics/Amgen, Reykjavik, Iceland. [9]Sleep Disorders Clinic, Department of Neurology, Medical University of Innsbruck, Innsbruck, Austria. [10]Sleep–Wake Disorders Center, Department of Neurology, Hôpital Gui-de-Chauliac, CHU Montpellier, Institut des Neurosciences de Montpellier, INSERM, Université de Montpellier, Montpellier, France. [11]SomnoDiagnostics, Osnabrück, Germany. [12]Department of Neurology, University Medical Center Göttingen, Göttingen, Germany. [13]Department of Neurology and Centre of Clinical Neuroscience, Charles University, First Faculty of Medicine and General University Hospital, Prague, Czech Republic. [14]Department of Neurology, Ludwig Maximilians University Munich, Munich, Germany. [15]Paracelsus-Elena-Klinik, Kassel, Germany. [16]Department of Neurosurgery, University Medical Center Göttingen, Göttingen, Germany. [17]Department of Neurology, Philipps-University Marburg, Marburg, Germany. [18]Neuropsychiatry Centre Erding/München, Erding, Germany. [19]Estonian Genome Center, Institute of Genomics, University of Tartu, Tartu, Estonia. [20]Department of Neurology, Nicosia General Hospital Medical School, University of Cyprus, Nicosia, Cyprus. [21]Bragée ME/CFS Center, Stockholm, Sweden. [22]Department of Pulmonology, Center of Sleep Medicine, Charité—Universitätsmedizin Berlin, Berlin, Germany. [23]Department of Neuroscience, Mayo Clinic College of Medicine, Jacksonville, FL, USA. [24]Department of Neurology, Mayo Clinic, Jacksonville, FL, USA. [25]Department of Clinical Medicine, University of Copenhagen, Copenhagen, Denmark. [26]Department of Clinical Immunology, Aarhus University Hospital, Aarhus, Denmark. [27]Department of Clinical Medicine, Aarhus University, Aarhus, Denmark. [28]Department of Clinical Immunology, Zealand University Hospital, Køge, Denmark. [29]Department of Clinical Immunology, Odense University Hospital, Odense, Denmark. [30]Department of Clinical Immunology, Aalborg University Hospital,

Aalborg, Denmark. [31]British Heart Foundation Cardiovascular Epidemiology Unit, Department of Public Health and Primary Care, University of Cambridge, Cambridge, UK. [32]British Heart Foundation Centre of Research Excellence, University of Cambridge, Cambridge, UK. [33]National Institute for Health and Care Research Blood and Transplant Research Unit in Donor Health and Behaviour, University of Cambridge, Cambridge, UK. [34]Health Data Research UK Cambridge, Wellcome Genome Campus and University of Cambridge, Cambridge, UK. [35]Victor Phillip Dahdaleh Heart and Lung Research Institute, University of Cambridge, Cambridge, UK. [36]Department of Haematology, University of Cambridge, Cambridge, UK. [37]Department of Human Genetics, the Wellcome Trust Sanger Institute, Wellcome Trust Genome Campus, Hinxton, UK. [38]NHS Blood and Transplant, Cambridge Biomedical Campus, Cambridge, UK. [39]Department of Haematology, University College London Hospitals, London, UK. [40]Radcliffe Department of Medicine and National Health Service Blood and Transplant, Oxford, UK. [41]Department of Haematology and BRC Haematology Theme, Churchill Hospital, Headington, Oxford, UK. [42]Magdalene College, Cambridge, UK. [43]23andMe, Inc., Sunnyvale, CA, USA. [44]Center for Restless Legs Syndrome, Department of Neurology, Johns Hopkins University, Baltimore, MD, USA. [45]Department of Neurology, Methodist Neurological Institute, Weill Cornell Medical School, Houston, TX, USA. [46]The Neuro (Montreal Neurological Institute–Hospital), McGill University, Montreal, Quebec, Canada. [47]Department of Neurology and Neurosurgery, McGill University, Montreal, Quebec, Canada. [48]Centre d'Études Avancées en Médecine du Sommeil, Hôpital du Sacré-Cœur de Montréal, Montreal, Quebec, Canada. [49]Department of Neurosciences, Université de Montréal, Montreal, Quebec, Canada. [50]Clinical and Molecular Metabolism Research Program (CAMM), Faculty of Medicine, University of Helsinki, Helsinki, Finland. [51]Department of Public Health and Welfare, National Institute for Health and Welfare, Helsinki, Finland. [52]Department of Molecular Biology of Cancer, Institute of Experimental Medicine, Academy of Science of Czech Republic, Prague, Czech Republic. [53]First Faculty of Medicine, Charles University in Prague, Prague, Czech Republic. [54]Biomedical Centre, Faculty of Medicine in Pilsen, Charles University in Prague, Pilsen, Czech Republic. [55]L'institut du thorax, CNRS, INSERM, Nantes Université, Nantes, France. [56]Department of Genetic Epidemiology, Institute for Human Genetics, University of Münster, Münster, Germany. [57]Institute of Clinical Molecular Biology, Kiel University, Kiel, Germany. [58]PopGen Biobank and Institute of Epidemiology, Christian Albrechts University Kiel, Kiel, Germany. [59]John and Jennifer Ruddy Canadian Cardiovascular Genetics Centre, University of Ottawa Heart Institute, Ottawa, Ontario, Canada. [60]Department of Medicine, Duke University School of Medicine, Durham, NC, USA. [61]Duke Clinical Research Institute, Duke University School of Medicine, Durham, NC, USA. [62]Institute of Epidemiology, Helmholtz Zentrum München, German Research Center for Environmental Health, Neuherberg, Germany. [63]Research Unit of Molecular Epidemiology, Helmholtz Zentrum München, German Research Center for Environmental Health, Neuherberg, Germany. [64]German Research Center for Cardiovascular Disease (DZHK), partner site Munich Heart Alliance, Hannover, Germany. [65]Chair of Epidemiology, Institute for Medical Information Processing, Biometry and Epidemiology, Medical Faculty, Ludwig-Maximilians-Universität München, Munich, Germany. [66]Department of Neurology, Emory University, Atlanta, GA, USA. [67]Department of Human Genetics, McGill University, Montreal, Quebec, Canada. [68]Institute of Epidemiology and Social Medicine, University of Münster, Münster, Germany. [69]Statens Serum Institute, Copenhagen, Denmark. [70]Health Data Science Research Centre, Fondazione Human Technopole, Milan, Italy. [71]Neurogenetic Systems Analysis Group, Institute of Neurogenomics, Helmholtz Zentrum München, German Research Center for Environmental Health, Neuherberg, Germany. [72]Munich Cluster for Systems Neurology (SyNergy), Munich, Germany. [73]German Center for Mental Health (DZPG), partner site Munich–Augsburg, Munich–Augsburg, Germany. [78]These authors contributed equally: Barbara Schormair, Chen Zhao, Steven Bell. [79]These authors jointly supervised this work: Emanuele Di Angelantonio, Konrad Oexle, Juliane Winkelmann. *Lists of authors and their affiliations appear at the end of the paper. ✉e-mail: barbara.schormair@helmholtz-munich.de

## 23andMe Research Team

**Nicholas A. Furlotte[43], Priyanka Nandakumar[43] & David A. Hinds[43]**

Full lists of members and their affiliations appear in the Supplementary Information.

## D.E.S.I.R. study group

**Amélie Bonnefond[74,75] & Louis Potier[76,77]**

[74]Inserm U1283, CNRS UMR 8199, European Genomic Institute for Diabetes, Institut Pasteur de Lille, Lille, France. [75]University of Lille, Lille University Hospital, Lille, France. [76]Institut Necker-Enfants Malades, INSERM UMR-S1151, CNRS UMR-S8253, Université Paris Cité, Paris, France. [77]Department of Diabetology, Endocrinology and Nutrition, DHU FIRE, Assistance Publique-Hôpitaux de Paris, Bichat Hospital, Paris, France.

## Methods

### Ethics statement

All studies were approved by the respective local ethical committees, and all participants provided informed consent. The EU-RLS-GENE study was approved by an institutional review board at the University Hospital of the Technical University of Munich (2488/09). The INTER-VAL dataset was approved by the National Research Ethics Service Committee East of England–Cambridge East (REC 11/EE/0538). Participants of 23andMe provided informed consent under a protocol approved by the external AAHRPP-accredited IRB, Ethical and Independent (E&I) Review Services. As of 2022, E&I Review Services is part of Salus IRB (https://www.versiticlinicaltrials.org/salusirb). The deCODE dataset was approved by the National Bioethics Committee of Iceland. The Danish Blood Donor Study (DBDS) dataset was approved by the Scientific Ethical Committee of Central Denmark (M-20090237) and by the Danish Data Protection agency (30-0444). GWAS studies in the DBDS were approved by the National Ethical Committee (NVK-1700407). The Emory dataset was approved by an institutional review board at Emory University, Atlanta, GA, USA (HIC ID 133-98).

### GWAS phenotyping and genotyping

Some of the samples were included already in our previous GWAS meta-analysis[3]. The reported sample numbers are the final sample numbers after quality control. Additional details are provided in the Supplementary Note.

**Discovery meta-analysis.** *International EU-RLS-GENE consortium (7,248 cases (2,479 males and 4,769 females) and 19,802 controls (10,422 males and 9,380 females)).* RLS cases were recruited in specialized outpatient clinics for movement disorders and in sleep clinics in European countries (Austria, Czech Republic, Estonia, Finland, France, Germany and Greece), Canada (Quebec) and the USA. RLS was diagnosed in a face-to-face interview by an expert neurologist or sleep specialist based on IRLSSG diagnostic criteria[1]. Controls were either population-based unscreened controls (Austria, Estonia, Finland, France, Germany) or healthy individuals recruited in hospitals (Canada, Czech Republic, Greece, USA). A total of 6,228 cases and 10,992 ancestry-matched controls had been genotyped on the Axiom array and were the study sample used in our previous meta-analysis. For the current study, 1,020 cases and 8,810 ancestry-matched controls were added who were genotyped on the Infinium Global Screening Array-24 version 1.0. Genotype calling was performed in GenomeStudio 2.0 according to the GenomeStudio Framework User Guide, and identical quality-control criteria were used for both datasets. Imputation was performed on the UK10K haplotype and 1000 Genomes Phase 3 reference panel using the EAGLE2 (version 2.0.5) and PBWT (version 3.1) imputation tools as implemented in the Sanger imputation server. Imputed SNPs with pHWE ≤ 1 × 10$^{-5}$ or an INFO score < 0.5 were filtered out.

*INTERVAL study (3,491 cases (1,291 males and 2,200 females) and 23,741 controls (12,511 males and 11,230 females)).* The INTERVAL study includes whole-blood donors recruited in England between 2012 and 2014. The Cambridge-Hopkins Restless Legs questionnaire was used to define RLS cases, and probable and definite cases were combined to form a binary phenotype as described previously[3]. A detailed description of Axiom 'Biobank' array genotyping and the imputation procedure plus related quality control in the INTERVAL trial can be found elsewhere[34]. Briefly, imputation was performed using a joint UK10K and 1,000 Genomes Phase 3 (May 2013 release) reference panel via the Sanger imputation server, and variants with MAF ≥ 0.1% and INFO score ≥ 0.4 were retained for analysis.

*Research participant cohort for 23andMe (105,908 cases (34,544 males and 71,364 females) and 1,502,923 controls (678,661 males and 824,262 females)).* This study includes research participants of 23andMe who

agreed to participate in research studies. The RLS phenotype was defined by self-reported responses to survey questions that assessed whether someone had ever been diagnosed with RLS or had ever received treatment for RLS as described previously[3]. Participants were genotyped on one of five platforms, all using Illumina arrays with added custom content (HumanHap550+ BeadChip, OmniExpress+ BeadChip, Infinium Global Screening Array). Participant genotype data were imputed in a two-step procedure using a reference panel created by combining the May 2015 release of the 1000 Genomes Phase 3 haplotypes with the UK10K imputation reference panel. Pre-phasing was carried out using either the internally developed tool Finch, which implements the Beagle algorithm, or EAGLE2. Imputation was performed with Minimac3.

**Replication meta-analysis.** *Research participant cohort for 23andMe (19,214 cases and 347,000 controls).* This cohort includes only individuals who had not been part of the 23andMe GWAS used in the discovery meta-analysis. Cases and controls were defined as described above.

*INTERVAL replication cohort (1,591 cases and 10,000 controls).* Individuals in this cohort do not overlap with samples included in the INTERVAL GWAS used in the discovery meta-analysis. RLS status was assessed with a single question on having received a diagnosis of RLS.

For 23andMe and INTERVAL, genotyping and imputation was carried out as described for the discovery stage.

*deCODE–DBDS–Emory cohort (8,223 cases and 41,815 controls).* This dataset included the DBDS, a cohort from deCODE Genetics, Iceland, the Emory Hospital Atlanta, USA and the Donor InSight-III study. Phenotyping and genotyping procedures have been described in detail previously[4].

### SNP-based association analysis

**Discovery-stage GWAS of autosomes.** *EU-RLS-GENE GWAS.* First, the Axiom- and the GSA-genotyped datasets were analyzed separately using SNPTEST version 2.5.4 with genotype dosages and assuming an additive model. Age, sex and the first ten PCs from the MDS analysis in PLINK were included as covariates. These summary statistics of the two datasets were then combined by fixed-effect inverse-variance meta-analysis (STERR scheme) using METAL (release 2011-03-25)[35]. One round of genomic control was performed in each dataset before meta-analysis.

*INTERVAL GWAS.* Assuming an additive genetic model, genotype dosages were analyzed in SAIGE (0.35.8.8) using a linear mixed model to account for cryptic relatedness and saddle point approximation to account for case–control imbalance[36]. Age, sex and the first ten PCs of ancestry were included as potential genomic confounders. The analysis was restricted to genetic variants with MAF ≥ 0.001, INFO ≥ 0.4 and a minor allele count of 10.

*The 23andMe GWAS.* Association analysis was conducted by logistic regression (LRT) assuming additive allelic effects and imputed dosages. Age, sex, genotyping platform and the first ten PCs were included as covariates.

In all individual GWAS, sex-specific analyses were performed using the same pipelines as those for the pooled analyses minus adjustment for sex as a covariate.

**Discovery-stage meta-analysis for autosomes.** We applied the same methods for both the pooled and the sex-specific GWAS. The three independent datasets were combined in a multivariate GWAS meta-analysis using the *N*-weighted-GWAMA R function (version 1.2.6)[37]. To assess the possibility of heterogeneity of SNP effects between the studies, Cochran's *Q*-test was applied as described in METAL.

**Discovery-stage meta-analysis for chromosome X.** Data for the X chromosome were available in two of the discovery-stage datasets: EU-RLS-GENE and 23andMe.

*EU-RLS-GENE XWAS.* For the pooled association analysis, male genotypes were coded as 0/2 (assuming no dosage compensation in males). All other methods were identical to those of the autosomal analyses. In sex-stratified analyses, males were coded as 0/1 and females as 0/1/2.

*The 23andMe XWAS.* In both pooled and sex-stratified analyses, males were coded as 0/2 and females as 0/1/2.

Pooled and sex-specific meta-analyses were performed using the N-GWAMA R function as in the autosomal analysis. Because N-GWAMA operates with $Z$ scores, the type of male allele coding did not affect the results.

**Sex-specific meta-analysis association analysis.** We performed sex-specific (male-only and female-only) meta-analyses of the corresponding GWAS using the N-GWAMA approach as described above. The results were used to estimate sex-specific heritability and genetic correlation between the sexes.

To detect sex-specific effects, we tested all independent ($r^2 < 0.2$) genome-wide significant SNPs of the pooled and sex-specific meta-analyses for heterogeneity of effect sizes between the two sexes using Cochran's $Q$-test (one-sided) and a Bonferroni-corrected significance threshold of $P_{adj} \leq 0.05/221$.

**Replication-stage association analysis.** For 23andMe and INTERVAL, quality control and statistical analysis were performed as described for the discovery stage. Statistical analysis for the DBDS, deCODE–Emory and Donor Insight studies has been described previously[4]. Meta-analysis was performed using Han and Eskin's random-effects model in METASOFT (RE2, METASOFT version 2.0.1)[38].

**Identification of risk loci and independent lead SNPs.** To define independent risk loci, we first used the '--clump' command in PLINK (version 1.90b6.7)[39] to collapse multiple genome-wide significant association signals based on linkage disequilibrium (LD) and distance (clump-r2 > 0.05, clump-kb < 500 kb clump-p1 < $5 \times 10^{-8}$, clump-p2p-value < $10^{-5}$). We then performed conditional analyses to identify secondary independent signals in risk loci using GCTA (version 1.93.0beta) with the '-cojo-slct' option, the $P$-value threshold for genome-wide significance set at $5 \times 10^{-8}$, the distance window set at 10 Mb and the colinearity cutoff set at 0.9 (ref. 40). LD was derived from EU-RLS-GENE genotype data. Independent genome-wide significant signals were merged into one genomic risk locus if either their LD block distance was <500 kb or their clumped regions were overlapping.

**Heritability analyses**
Heritability is reported on the liability scale unless otherwise indicated. Prevalence estimates were derived from the population cohorts INTERVAL and 23andMe themselves. For the EU-RLS-GENE case–control dataset and for the meta-analysis, prevalence estimates were derived from previous publications on European ancestries.

We estimated SNP-based heritability under several different heritability models. LDSC (version 1.0.1) was used with standard settings, invoking a model where SNPs with different MAFs are expected to contribute equally to heritability[41]. LDAK (version 5.0) was used with standard settings to implement the LDAK model, where SNP contributions depend on LD structure and MAF as well as the BLD-LDAK and BLD-LDAK+Alpha models, which incorporate additional annotation-based features[42]. All analyses were based on summary statistics and filtering according to LDSC default settings, that is,

HapMap3 non-HLA SNPs with MAF > 0.01 and INFO ≥ 0.9. The Akaike information criterion of each of these models was reported for model comparison. Further details are provided in the Supplementary Note.

For X chromosome heritability estimation, we followed the approach described by Lee et al. and used the summary statistics of the N-GWAMA meta-analysis[43]. For sex $k$, the SNP heritability $h_k^2$ relates to the expected $\chi^2$ statistics as $\mathbb{E}(\chi_k^2) \approx 1 + N_k h_k^2 / M_{eff}$, where $N_k$ is the GWAS sample size, and $M_{eff}$ is the effective number of loci within the examined genomic region (assumed to be the same in males and females). For calculation of the (sex-specific) relative heritability contribution of the X chromosome, $\chi^2$ statistic-based $h^2$ was also calculated for the autosomes.

**Genetic correlation analysis**
For autosomal data, genetic correlations were calculated using LDSC (version 1.0.1) using the same SNP filtering criteria and the two-step estimation option as in the heritability estimation. Because the LDSC framework is not applicable for chromosome X, the genetic correlation coefficient $\hat{r}_g$ was estimated as $\hat{r}_g = \frac{\overline{Z_m Z_f}}{\sqrt{(\hat{\chi}_f^2 - 1)(\hat{\chi}_m^2 - 1)}}$, where $Z$ and $\chi^2$ are the $Z$ scores and mean $\chi^2$ estimates from the female (f) and male (m)-specific studies.

In addition to between-study and between-sex genetic correlations, we performed a large-scale genetic correlation screen for RLS (represented by the pooled autosomal meta-analysis data) and other traits using LDSC as described above. Sources and filtering criteria for summary statistics included in this screen are provided in the Supplementary Note.

Traits significantly correlated with RLS (FDR < 0.05, one-sample two-sided $Z$-test) were taken forward to a bi-serial genetic correlation analysis. Here, we computed the pairwise $\hat{r}_g$ between all traits.

An unsigned weighted correlation matrix was built using the pairwise $\hat{r}_g$ and used as input for a weighted correlation matrix analysis to perform hierarchical clustering and to detect modules with the WGCNA package (version 1.69)[44]. The following settings were applied in WGCNA: softPower, 6; network type, 'unsigned'; TOMDenom, 'min'; Dynamic-cutree, method = 'hybrid'; deepSplit, 2; minModuleSize, 30; pamStage, TRUE; pamRespectsDendro, FALSE; useMedoids, FALSE. The defining trait categories in each module were determined by consensus through independent review of the within-module cluster structure by visual inspection of network plots at two sites (Helmholtz and Cambridge).

**Mendelian randomization**
To select traits for MR, we defined two to eight clusters in a module based on its complexity. In each cluster, the traits were ranked according to the significance of their correlation with RLS, and we selected the most significantly correlated medical conditions or potentially modifiable lifestyle factors. We supplemented this list with traits for which an association with RLS has been described in the literature.

Using R version 4.0.4, we filtered GWAS datasets to uncorrelated SNPs ($r^2 < 0.01$ in the European 1000 Genomes Phase 3 data), aligned them to GRCh37 and mapped them to dbSNP 153 with the gwasvcf package (version 0.1.0). We harmonized effect alleles across studies using the TwoSampleMR package (version 0.5.6)[45]. Palindromic variants with ambiguous allele frequencies and those with unresolved strand issues were excluded from analysis.

To avoid violations of the classical MR assumptions when studying correlated and likely pleiotropic traits, we used a robust method for bidirectional MR, LHC-MR (version 0.0.0.9000)[32]. Traits with low heritability ($h^2 < 2.5\%$, $P_{h^2} > 0.05$) were excluded from the analysis. Significance of directionality and confounding effect were tested by comparing the goodness of fit of six degenerate LHC-MR models (only latent effect, only causal effect, only causal effect to RLS, only causal effect from RLS, no causal effect to RLS and no causal effect from RLS)

to the full model. We supplemented these analyses with those based on the IVW and MR-Egger methods.

## Gene prioritization in risk loci

All analyses were performed on the N-GWAMA results of the pooled meta-analysis. We applied several complementary approaches to prioritize candidate genes in the genome-wide significant risk loci. These included the gene-prioritization pipeline of DEPICT (version 1.rel194), three prioritization workflows (positional, eQTL-based and topology-based mapping) provided on the FUMA platform (https://fuma.ctglab.nl/, version 1.3.6a), a gene-level GWAS using MAGMA version 1.08, a transcriptome-wide association study using S-PrediXcan and S-MultiXcan (MetaXcan package version 0.7.4), a colocalization analysis with eCAVIAR (version 2.2) and statistical fine-mapping with CAVIARBF (version 0.2.1)[46–52]. In the DEPICT, FUMA eQTL-based mapping, MAGMA and transcriptome-wide association study analyses, a gene was considered prioritized if it had an FDR < 0.05; in FUMA topology-based mapping, if it had an FDR < $1 \times 10^{-5}$; and in eCAVIAR, if it had a colocalization posterior probability > 0.1. In FUMA positional mapping, a gene was considered prioritized if genome-wide significant SNPs physically mapped to it. In statistical fine-mapping, a gene was considered prioritized if an SNP in the 95% credible set of the risk locus could be linked to it by either eQTL, chromatin interaction or positional mapping. In addition, we checked whether a gene contained genome-wide significant coding variants (the gene was considered prioritized if it did) and whether a gene mapped to a gene set that was significant in our enrichment analyses (the gene was considered prioritized if it did). We combined the results of all approaches per gene in a prioritization score by summing up the individual results, counting 'not prioritized' as 0 and 'prioritized' as 1. Further details are provided in the Supplementary Note.

## Enrichment analyses

**Gene set and pathway enrichment analyses.** *DEPICT*. We ran DEPICT to detect enrichment of gene sets across risk loci as well as to identify tissue and cell types where expression is enriched for genes across risk loci. We set the significance thresholds for lead SNPs at $1 \times 10^{-5}$ and at $5 \times 10^{-4}$ for null GWAS; all other settings were the same as those used for gene prioritization (see above). DEPICT was run with all built-in datasets. eQTL mapping and functional prioritization were evaluated in DEPICT's built-in eQTL and reconstituted gene sets.

Excluding 12 SNPs not reaching genome-wide significance in the joint analysis of discovery and validation did not change the main results (Supplementary Table 25).

*MAGMA*. MAGMA (version 1.08) was used to perform gene set enrichment testing for pathway identification. MAGMA conducts competitive gene set tests with correction for gene size, variant density and LD structure. A total of 7,522 gene sets representing the GO biological process ontology (MSigDB version 7.1, C5 collection, GO:BP subset) were tested for association. We adopted a significance threshold of FDR < 0.05 (one-sided *t*-test).

**Tissue and cell type enrichment analyses.** Using the settings described above, we tested enrichment of RLS heritability with DEPICT across 209 different tissue types covered in the built-in dataset. For an independent validation on the tissue level as well as for the analyses on the cell type level, we mainly used the CELLEX and CELLECT tools[53]. CELLECT provides two different gene-prioritization approaches for heritability enrichment testing, S-LDSC and MAGMA covariate analysis[54,55]. For compatibility of the results, the summary statistics of the pooled N-GWAMA analysis were filtered using settings identical to those in our LDSC heritability analyses. Following the recommendations by Timshel et al.[53], we applied a 'tiered' approach by starting with body-wide datasets and then focusing on CNS-centric datasets. We used

CELLECT software (version 1.3.0) with default settings but updated to MAGMA version 1.08 to test enrichment of RLS heritability in cell type- or tissue-specific genes for datasets with publicly available RNA-seq data. These analyses require a measure of expression specificity for each gene in a cell or tissue type. We either used CELLEX (version 1.2.1) to compute expression specificity or relied on precomputed CELLEX expression specificity scores. Human adult datasets without publicly available raw RNA-seq data were analyzed using MAGMA_Celltyping (version 2.0.0) in top10 mode. The list of input datasets is provided in the Supplementary Note, and results of our evaluation of both approaches showing high correlation are presented in Supplementary Fig. 1 and Supplementary Table 26.

## Risk prediction

We applied three types of models for genetic risk evaluation and RLS risk prediction: GLM with and without interaction terms, RF models and DNN models. These were implemented as binary classifiers as well as time-to-event classifiers.

Training of the models and evaluation by tenfold cross-validation were based on the EU-RLS-GENE Axiom subset. Therefore, we first conducted a meta-analysis excluding this dataset to generate unbiased summary statistics to be used in all models. Because GWAS have an ascertainment bias, we constructed a simulation cohort dataset by resampling of the EU-RLS-GENE Axiom subset based on the year of birth of the sampled individuals, their ages at onset and the demographic composition of the German population (Supplementary Note). We calculated the PRS using dosages of 216 independent lead SNPs of our discovery meta-analyses.

For a baseline comparison of the predictive power of this score to a PRS based on genome-wide data, we calculated a genome-wide PRS using the LDpred2-auto option of LDpred2 (R package bigsnpr version 1.12.2)[56]. Variants and the LD reference panel were based on the HapMap3 EUR dataset, and window size for calculating SNP correlation was set to 3 cM.

Binary classification models were evaluated by Nagelkerke's pseudo-$R^2$, receiver operator characteristic AUC and precision–recall AUC. A 5-year binary classifier was constructed for each of the time-to-event models by predicting the label until the next 5 years and evaluated by the metrics for binary classification.

To evaluate the contribution of the interaction effects to model performance, we estimated the effect sizes of interaction terms such as PRS × age by logistic regression:

$$P(\text{RLS} = 1|\text{PRS, sex, age, } \textbf{PC})$$

$$= \frac{1}{1 + e^{-(\beta_0 + \beta_1 \text{PRS} + \beta_2 \text{sex} + \beta_3 \text{age} + \beta_4 \text{PRS} \times \text{sex} + \beta_5 \text{PRS} \times \text{age} + \beta_6 \text{sex} \times \text{age} + \beta_7 \text{PRS} \times \text{sex} \times \text{age} + \gamma \cdot \textbf{PC})}},$$

where age is the dummy variable of age in bins of 20 years, **PC** indicates the first ten PCs from the MDS analysis in PLINK, $\gamma$ is a vector of effect sizes of PCs and the PRS = $\Sigma_j w_j g_j$, where $w_j$ and $g_j$ are the per-allele effect size and dosage of the *j*-th SNP, respectively.

For the DNN and RF models, we used these logistic regression estimates as the baseline and then further estimated the interaction effect sizes indirectly by calculating the incremental gain in explained variance (Nagelkerke's pseudo-$R^2$) from $\text{model}_0$ to $\text{model}_1$ as:

$$R^2 = \left(1 - (L(\text{model}_0)/L(\text{model}_1))^{\frac{2}{N}}\right)\left(1 - L(\text{model}_0)^{\frac{2}{N}}\right)^{-1},$$

where $L$ is the likelihood function for a logistic regression model with the first ten PCs included as covariates.

Binary classification models, GLMs and RF and DNN models were built, optimized and trained by H2O AutoML (version 3.36.0.2) in R (version 4.0.2)[57]. Time-to-event models were implemented with randomForestSRC (version 3.0.1) in R (version 4.0.2)

and PyTorch[58] (pycox version 0.2.1 and PyTorch version 1.6.0). Cross-validation-based Nagelkerke's pseudo-$R^2$ was calculated in R version 4.0.2.

## Reporting summary

Further information on research design is available in the Nature Portfolio Reporting Summary linked to this article.

## Data availability

Summary statistics of the meta-analysis are publicly available for the top 10,000 SNPs at Zenodo (https://doi.org/10.5281/zenodo.10804907)[59]. Summary statistics of the discovery-stage International EU-RLS-GENE consortium GWAS and the INTERVAL GWAS are available at the GWAS Catalog (https://www.ebi.ac.uk/gwas/) under accession codes GCST90399568, GCST90399569, GCST90399570, GCST90399571, GCST90399572 and GCST90399573. The full GWAS summary statistics for the 23andMe discovery dataset have been made available through 23andMe to qualified researchers under an agreement with 23andMe that protects the privacy of the 23andMe participants. Datasets have been made available at no cost for academic use. Please visit https://research.23andme.com/collaborate/#dataset-access/ for more information and to apply to access the data. Additional data used for tissue and cell type enrichment analysis are available here: developmental (http://mousebrain.org/development/downloads.html) and adult single-cell RNA-seq datasets (http://mousebrain.org/adult/downloads.html) from the Mouse Brain Atlas (http://mousebrain.org/), the Human Gene Expression During Development dataset from the BBI-Allen Single Cell atlases (https://descartes.brotmanbaty.org/), the BrainSpan Developmental Transcriptome RNA-seq dataset from the BrainSpan Atlas of the Developing Human Brain (https://www.brainspan.org/static/home), the V8 RNA-seq dataset (GTEx_Analysis_2017-06-05_v8_RNASeQCv1.1.9_gene_reads.gct.gz) from GTEx (https://gtexportal.org/home/datasets) and the human C8 collection from MSigDb version 7.4 (http://software.broadinstitute.org/gsea/msigdb/), with legacy versions available at https://www.gsea-msigdb.org/gsea/downloads_archive.jsp after creating a user account with GSEA–MSigDB. Summary statistics of GWAS for genetic correlation and MR analyses are available at the University of Bristol Integrative Epidemiology Unit OpenGWAS server (https://gwas.mrcieu.ac.uk) and the GWAS Atlas (https://atlas.ctglab.nl/). Additional GWAS summary statistics for iron-related traits are available at https://www.fmrib.ox.ac.uk/ukbiobank/gwas_resources/index.html, https://open.win.ox.ac.uk/ukbiobank/big40/BIGv2/ and https://www.decode.com/summarydata/. A complete list of sources used for annotation with FUMA is available at https://fuma.ctglab.nl/links and https://fuma.ctglab.nl/tutorial. Auxiliary files for use with MAGMA are available at https://ctg.cncr.nl/software/magma. Additional files for use with LDSC and LDAK are available at https://alkesgroup.broadinstitute.org/LDSCORE/. Information about drug targets is available at the free-to-access database DrugBank Online (https://go.drugbank.com/).

## Code availability

We provide information on publicly available software and settings in the Methods and the Supplementary Note. For custom data analysis, we describe the theoretical background as well as the models used in detail in the Supplementary Note. Custom code scripts are available on Zenodo (https://doi.org/10.5281/zenodo.10804907)[59]. Publicly available software used in this study includes PLINK (version 1.90b6.7, https://www.cog-genomics.org/plink/1.9/), SNPTEST (version 2.5.4, https://www.chg.ox.ac.uk/~gav/snptest/), SAIGE (0.35.8.8, https://github.com/saigegit/SAIGE), N-GWAMA (version 1.2.6, https://github.com/baselmans/multivariate_GWAMA), METAL (release 2011-03-25, https://csg.sph.umich.edu/abecasis/metal/index.html), METASOFT (version 2.0.1, https://web.cs.ucla.edu/~eeskin/), GCTA (version 1.93.0beta, https://yanglab.westlake.edu.cn/software/gcta/#Overview), LDSC

(version 1.0.1, https://github.com/bulik/ldsc), LDAK (version 5.0, https://dougspeed.com/ldak/), LHC-MR (version 0.0.0.9000, https://github.com/LizaDarrous/lhcMR), DEPICT (version 1 rel194, https://github.com/perslab/depict), FUMA (version 1.3.6a, https://fuma.ctglab.nl/), MAGMA (version 1.08, https://cncr.nl/research/magma/), MetaXcan (version 0.7.4, https://github.com/hakyimlab/MetaXcan), eCAVIAR (version 2.2, https://github.com/fhormoz/caviar), CAVIARBF (version 0.2.1, https://bitbucket.org/Wenan/caviarbf/src/master/), CELLECT (version 1.3.0) and CELLEX (version 1.2.1) (https://github.com/perslab/CELLECT), MAGMA_Celltyping (version 2.0.0, https://github.com/neurogenomics/MAGMA_Celltyping), pycox (version 0.2.1, https://github.com/havakv/pycox), PyTorch (version 1.6.0, https://github.com/pytorch/pytorch), H2O AutoML (version 3.36.0.2, https://docs.h2o.ai/h2o/latest-stable/h2o-docs/automl.html), the Sanger imputation server (https://imputation.sanger.ac.uk/), EAGLE2 (version 2.0.5, https://alkesgroup.broadinstitute.org/Eagle/), PBWT (version 3.1, https://github.com/richarddurbin/pbwt), Minimac3 (https://genome.sph.umich.edu/wiki/Minimac3), R (version 4.0.4 and version 4.0.2, https://cran.r-project.org/), the rrvgo package (version 1.2.0, https://bioconductor.org/packages/release/bioc/html/rrvgo.html), the WGCNA package (version 1.69, https://cran.r-project.org/web/packages/WGCNA/index.html), the TwoSampleMR package (version 0.5.6, https://mrcieu.github.io/TwoSampleMR/index.html), the coloc package (version 5.1.0, https://chr1swallace.github.io/coloc/), the BayesianTools package (version 0.0.10, https://github.com/florianhartig/BayesianTools), the bigsnpr package (version 1.12.2), implements LDpred2 (https://privefl.github.io/bigsnpr/), the gwasvcf package (version 0.1.0, https://github.com/MRCIEU/gwasvcf) and the randomForestSRC package (version 3.0.1, https://www.randomforestsrc.org/).

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

## Acknowledgements

For the International EU-RLS-GENE consortium, we thank all colleagues and staff at the participating centers of the International EU-RLS-GENE consortium for their help in recruiting study participants. We thank the German Restless Legs Syndrome Foundation (RLS e.V.) for continuously supporting our studies. We acknowledge the technical support of Core Facility Genomics at Helmholtz Zentrum München for part of the genotyping done for the EU-RLS-GENE GWAS. Funding was received from the Deutsche Forschungsgemeinschaft (grants 218143125 and 310572679 to J.W. and partial support to J.W. within SyNergy EXC 2145 grant 390857198); the European Regional Development Fund (project GenTransMed 2014-2020.4.01.15-0012 to M.T.-L. and A.M.); the University of Thessaly (grant 2845 to G.M.H.); the NIH–NIA and the NIH–NINDS (1U19AG063911, FAIN U19AG063911 to Z.K.W.); the Mayo Clinic Center for Regenerative Medicine, gifts from the Donald G. and Jodi P. Heeringa Family, the Haworth Family Professorship in Neurodegenerative Diseases fund and the Albertson Parkinson's Research Foundation (all to Z.K.W.); the National Institutes of Health National Institute of Neurological Disorders and Stroke (P50 NS072187 to O.A.R.); the Mayo Clinic Neuroscience Focused

Research Team (Cecilia and Dan Carmichael Family Foundation and the James C. and Sarah K. Kennedy Fund for Neurodegenerative Disease Research) (all to O.A.R.); the Canadian Institutes of Health Research (376503 to A.F.R.S.); the Natural Sciences and Engineering Research Council of Canada (RGPIN-2016-04985 to A.F.R.S.), the Canadian Diabetes Association (OG-3-14-4567-HC to A.F.R.S.), the Heart and Stroke Foundation of Canada (G-16-00014085 to A.F.R.S.); the Charles University Cooperation Program in Neuroscience and the Program EXCELES (LX22NPO5107 to D.K. and K. Sonka). Collection of samples by Emory University investigators was funded by the RLS and AL Williams Jr. Family Foundations. The PROCAM-2 Study was initiated and conducted by the Leibniz Institute for Arteriosclerosis Research at the University of Münster under the leadership of G. Assmann. After his retirement, all data were transferred to the university for further scientific use. The later follow-up assessment was carried out with funds from the Institute of Epidemiology and Social Medicine, and DNA isolation was performed with financial support from the Dean of the medical faculty, both at the University of Münster. Genotyping was enabled through funds from the German Center for Cardiovascular Disease. The Course of RLS Study (COR) was supported by unrestricted grants to the University of Münster from the German Restless Legs Society (RLS e.V. Deutsche Restless Legs Vereinigung) and Boehringer Ingelheim Pharma, Mundipharma Research, Roche Pharma, NeuroBioTec and UCB (Schwarz Pharma). The KORA study was initiated and financed by the Helmholtz Zentrum München—German Research Center for Environmental Health, which is funded by the German Federal Ministry of Education and Research (BMBF) and by the state of Bavaria. Data collection in the KORA study is done in cooperation with the University Hospital of Augsburg. Furthermore, KORA research was supported within the Munich Center of Health Sciences, Ludwig-Maximilians-Universität, as part of LMUinnovativ. We thank all participants for their long-term commitment to the KORA study, the staff for data collection and research data management and the members of the KORA Study Group (https://www.helmholtz-munich.de/en/epi/cohort/kora) who are responsible for the design and conduct of the study. For the INTERVAL study, we thank the NIH Research Cambridge Biomedical Research Centre for funding (RG64219). Participants in the INTERVAL randomized controlled trial were recruited with the active collaboration of NHS Blood and Transplant England (https://www.nhsbt.nhs.uk), which has supported field work and other elements of the trial. DNA extraction and genotyping were cofunded by the National Institute for Health and Care Research (NIHR), the NIHR BioResource (http://bioresource.nihr.ac.uk) and the NIHR Cambridge Biomedical Research Centre (BRC-1215-20014) (the views expressed in this paper are those of the author(s) and not necessarily those of the NIHR, NHSBT or the Department of Health and Social Care). The academic coordinating center for INTERVAL was supported by core funding from the NIHR Blood and Transplant Research Unit in Donor Health and Genomics (NIHR BTRU-2014-10024), the NIHR BTRU in Donor Health and Behaviour (NIHR203337), the UK Medical Research Council (MR/L003120/1), the British Heart Foundation (SP/09/002; RG/13/13/30194; RG/18/13/33946) and NIHR Cambridge BRC (BRC-1215-20014). A complete list of the investigators and contributors to the INTERVAL trial is provided in ref. 60. The academic coordinating center thanks blood donor center staff and blood donors for participating in the INTERVAL trial. This work was also supported by Health Data Research UK, which is funded by the UK Medical Research Council, the Engineering and Physical Sciences Research Council, the Economic and Social Research Council, the Department of Health and Social Care (England), the Chief Scientist Office of the Scottish Government Health and Social Care Directorates, the Health and Social Care Research and Development Division (Welsh Government), the Public Health Agency (Northern Ireland), the British Heart Foundation and Wellcome. Regarding the INTERVAL data, this

work was performed using resources provided by the Cambridge Service for Data Driven Discovery (CSD3) operated by the University of Cambridge Research Computing Service (https://www.csd3.cam.ac.uk), provided by Dell EMC and Intel using tier 2 funding from the Engineering and Physical Sciences Research Council (capital grant EP/P020259/1), and DiRAC funding from the Science and Technology Facilities Council (https://www.dirac.ac.uk). S.B. is supported by Cancer Research UK (A27657). W.H. Ouwehand is supported by grants to his laboratory from the National Institute for Health Research (NIHR), the European Commission (HEALTH-F2-2012-279233), the British Heart Foundation (RP-PG-0310-1002 and RG/09/12/28096) and NHSBT, and he is a senior investigator for the NIHR. N. Soranzo is supported by the Wellcome Trust (WT098051 and WT091310) and the European Commission Framework Programme 7 (EPIGENESYS 257082 and BLUEPRINT HEALTH-F5-2011-282510). D.J.R. is supported by the NIHR (NIHR-RP-PG-0310-1004). J. Danesh holds a British Heart Foundation Professorship and an NIHR Senior Investigator Award. P.V. was supported by LX22 NPO 5102. We also thank the DBDS and the DBDS Genomic Consortium for their contribution. A complete list of the investigators and contributors to the DBDS Genomic Consortium is provided in the Supplementary Note. The DBDS is funded by the Danish Council for Independent Research—Medical Sciences (8020-00403B), the Danish Administrative Regions and Bio- and Genome Bank Denmark, the Danish Blood Donor Research Foundation and the Novo Nordisk Foundation Challenge Program (NNF17OC0027594). We thank the research participants and employees of 23andMe for making this work possible. The 23andMe Research Team provided infrastructure for generating 23andMe data. Participants provided informed consent and volunteered to participate in the research online under a protocol approved by the external AAHRPP-accredited IRB, E&I Review Services. As of 2022, E&I Review Services is part of Salus IRB (https://www.versiticlinicaltrials.org/salusirb).

## Author contributions

B.S. and J.W. designed and coordinated the study. C.Z., S.B. and K.O. proposed and performed, contributed to and supervised bioinformatic procedures, statistical tests and meta-analyses, respectively. K.O., E.D.A. and J.W. supervised the overall study. B.S., C.Z., S.B., M.D., J. Dowsett., M.S.N., N.A.F., P.N. and D.A.H. performed statistical analysis within cohorts. N. Schandra, A.S., B.H., Y.D., C.G.B., D.K., K. Sonka, W.P., C.T., W.H. Oertel, M.H., G.M.H., O.P., I.F., J.W., O.A.R., Z.K.W., A.I., M.B., S.C., C.J.E., W.G.O., M.T.-L., A.M., A.D., L.X., G.A.R., K.B., M.P., P.V., C.D., A.F., W.L., A.F.R.S., S.H.S., C.G., A.P., M.S., the D.E.S.I.R. study group, the 23andMe Research Team, A.S.B., N. Soranzo, W.H. Ouwehand, D.J.R., J. Danesh, B.B., E.D.A., S.R.O., E.S., C.E., O.B.P., M.T.B., K.R.N., H.U., D.B.R., H.S. and K. Stefansson acquired data and samples within cohorts. V.K., P.H., N. Schandra, A.S., B.H., Y.D., C.G.B., D.K., K. Sonka, W.P., C.T., W.H. Oertel, M.H., G.M.H., O.P., I.F., J.W., O.A.R., Z.K.W., A.I., M.B., S.C., C.J.E., W.G.O., M.T.-L., A.M., A.D., L.X., G.A.R., K.B., M.P., P.V., C.D., A.F., W.L., A.F.R.S., S.H.S., C.G., A.P., M.S., the D.E.S.I.R. study group, the 23andMe Research Team, A.S.B., N. Soranzo, W.H. Ouwehand, D.J.R., J. Danesh, B.B., E.D.A., S.R.O., E.S., C.E., O.B.P., M.T.B., K.R.N., H.U., D.B.R., H.S., K. Stefansson and K.O. supported data interpretation within cohorts. B.S., C.Z., S.B., K.O., E.D.A. and J.W. performed data interpretation of meta-analyses. B.S., K.O., C.Z., S.B., J.W. and E.D.A. wrote the paper. B.S., K.O., C.Z., S.B., J.W., E.D.A., V.K., P.H., M.D., J. Dowsett, M.S.N., N.A.F., P.N., D.A.H., N. Schandra, A.S., B.H., Y.D., C.G.B., D.K., K. Sonka, W.P., C.T., W.H. Oertel, M.H., G.M.H., O.P., I.F., O.A.R., Z.K.W., A.I., M.B., S.C., C.J.E., W.G.O., M.T.-L., A.M., A.D., L.X., G.A.R., K.B., M.P., P.V., C.D., A.F., W.L., A.F.R.S., S.H.S., C.G., A.P., M.S., the D.E.S.I.R. study group, the 23andMe Research Team, A.S.B., N. Soranzo, W.H. Ouwehand, D.J.R., J. Danesh, B.B., S.R.O., E.S., C.E., O.B.P., M.T.B., K.R.N., H.U., D.B.R., H.S. and K. Stefansson reviewed and approved the final version of the paper.

## Funding

## Competing interests

The funders of the study had no role in conceptualization, design, data collection, analysis, the decision to publish or preparation of the manuscript. J.W., B.S., K.O. and C.Z. have filed a patent application (WO2021185936A1). Z.K.W. serves as PI or co-PI on Biohaven Pharmaceuticals (BHV4157-206), Neuraly (NLY01-PD-1) and Vigil Neuroscience (VGL101-01.002, VGL101-01.201, PET tracer development protocol, and CSF1R biomarker and repository project) grants. Z.K.W. serves as co-PI of the Mayo Clinic APDA Center for Advanced Research and as an external advisory board member for Vigil Neuroscience. W.P. has received honoraria as a speaker from Philips and MediPark Clinic and as a consultant from Abbott and Precisis. J. Danesh serves on scientific advisory boards for AstraZeneca, Novartis and the UK Biobank and has received multiple grants from academic, charitable and industry sources outside of the submitted work. A.S.B. reports institutional grants from AstraZeneca, Bayer, Biogen, BioMarin, Bioverativ, Novartis, Regeneron and Sanofi. D.A.H., N.A.F., P.N. and members of the 23andMe Research Team are employed by and hold stock or stock options in 23andMe. Authors affiliated with deCODE Genetics/Amgen declare competing financial interests as employees. The other authors declare no competing interests.

## Additional information

**Extended data** is available for this paper at https://doi.org/10.1038/s41588-024-01763-1.

**Correspondence and requests for materials** should be addressed to Barbara Schormair.

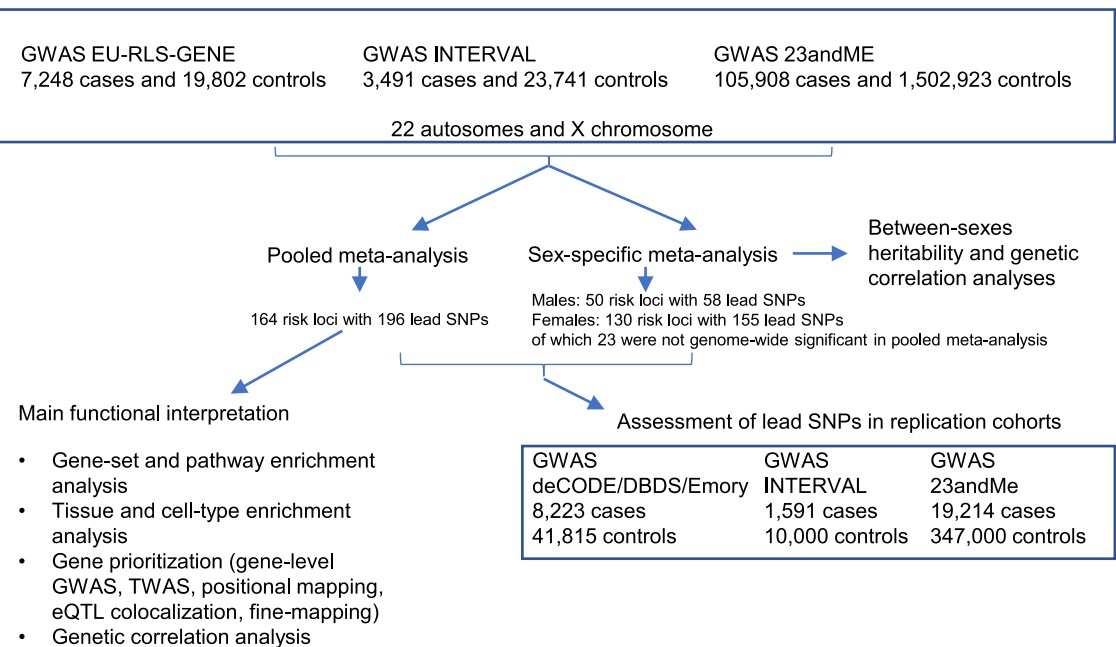

**Extended Data Fig. 1 | General study workflow.** Overview of the main analytical steps conducted in the study. While sex-specific GWAS meta-analysis results were used to dissect similarities and differences between both sexes, the pooled meta-analysis results were used for further functional interpretation.

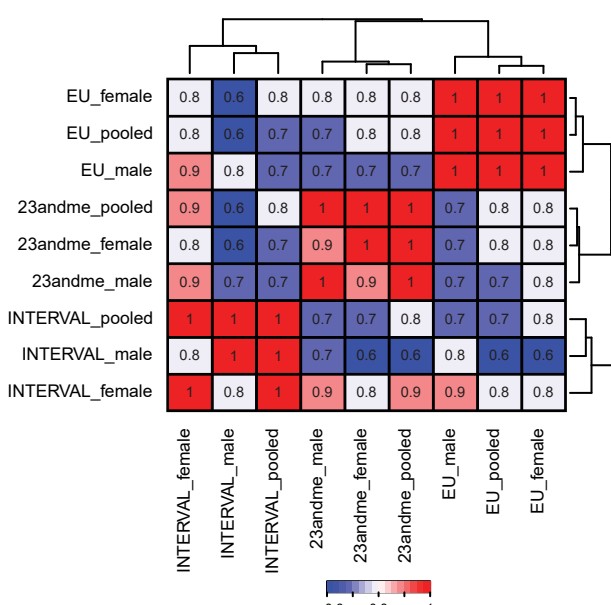

**Extended Data Fig. 2 | Genetic correlation between individual discovery stage GWAS of the N-GWAMA meta-analysis.** Genetic correlations between the discovery stage input GWAS were calculated using LDSC on the summary statistics.

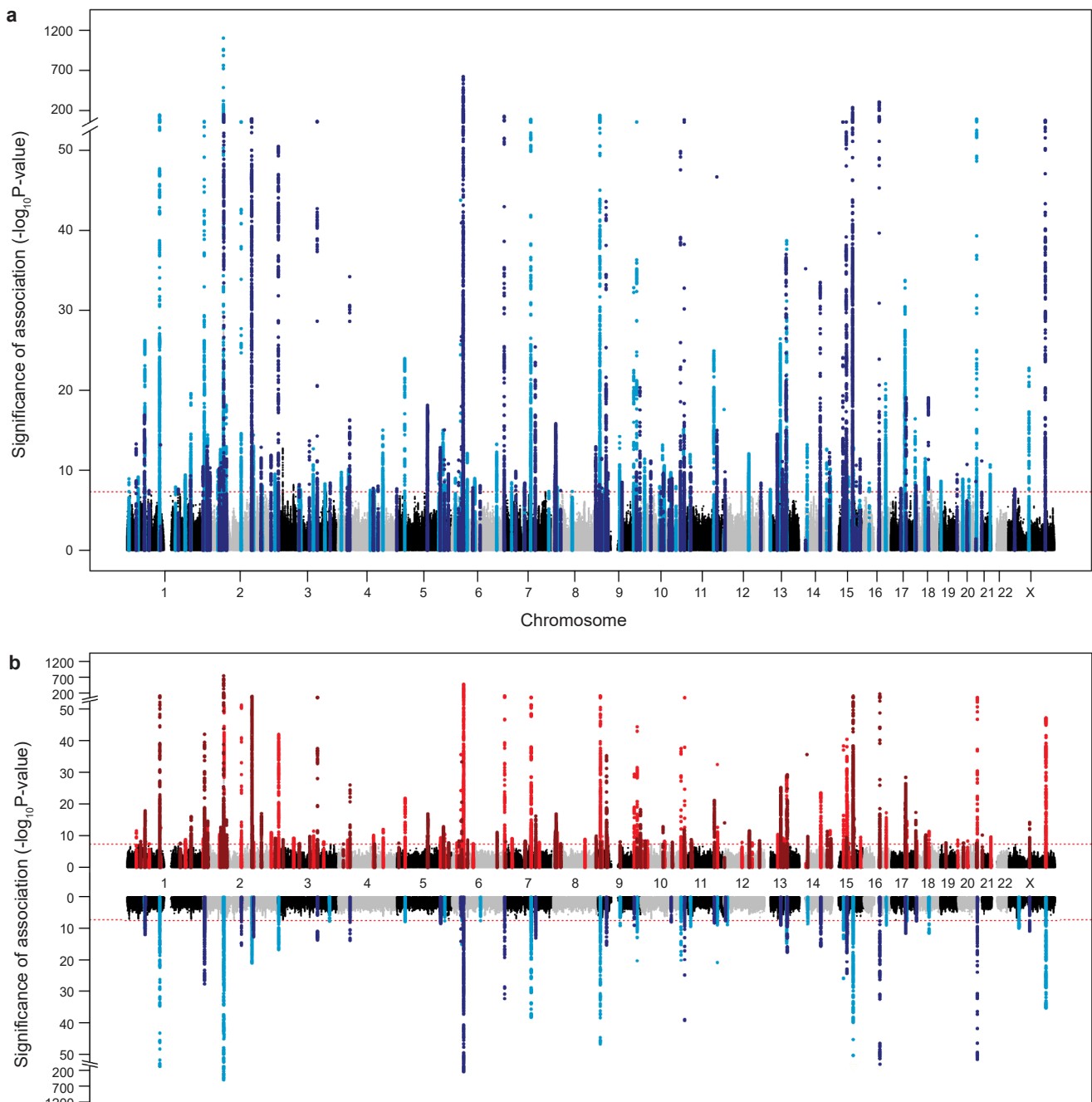

**Extended Data Fig. 3 | Manhattan and Miami plots of discovery stage meta-analyses. a**, Results of the pooled discovery meta-analysis. **b**, Results of the sex-specific discovery meta-analyses. Female-only results are depicted in red in the upper section of the Miami plot, male-only results are depicted in blue in lower section of the Miami plot. The x-axis shows chromosome and base pair positions of the tested variants. The y-axis shows significance as −log$_{10}$ of the two-sided nominal P-values of the N-GWAMA analyses. Red horizontal dashed lines indicate the Bonferroni-adjusted significant threshold of $P < 5 \times 10^{-8}$.

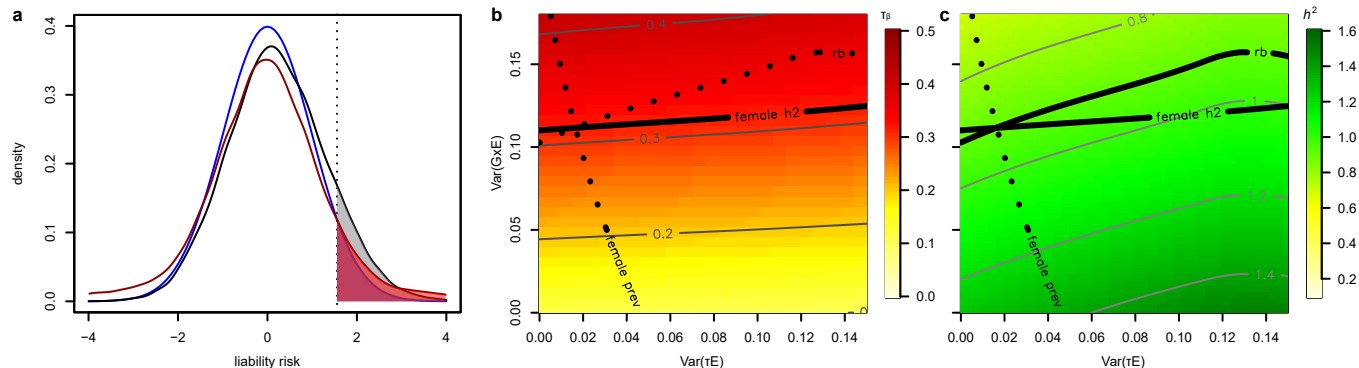

**Extended Data Fig. 4 | Simulation study assessing sex-specific heritability and genetic correlation divergence.** Simulation of environmental effect that reconciles sex-difference in heritability with the similarity of the SNP effect sizes. **a**, Frequency density distributions of the liabilities for different models. Blue line, base model, $\varphi = X\beta + \varepsilon$, as assumed to be present in males with $h^2 = 0.1395$, $X$ and $\beta$ as determined by GWAS, $\varepsilon \sim N(0, 1)$, and a disease threshold in keeping with the male RLS prevalence of 0.06 (shaded area under the curve). Black line, model with non-interacting binary environmental effect, $\varphi = X\beta + \tau E + \varepsilon$, with $X, \beta, \varepsilon$ and threshold as in the base model plus an additional binary effect $E \sim Bernoulli(p = 0.21)$, representing childlessness with a weight vector $\tau$ such that that prevalence is 0.13 as in females. Red line, analogous $G{\times}E$ model,

$\varphi = X\beta + X\eta \circ E + \varepsilon$, but where the environmental effect now interacts with the genetic effects and the corresponding weight vector $\eta$ is chosen in accordance with the female prevalence. **b**, **c**, Optimization of the model $\varphi = X\beta + X\eta \circ E + \tau E + \varepsilon$ with $X, \beta, E, \varepsilon$ and threshold as above, where the additional degree of freedom is covered by also considering the mean effect size ratio $rb$ observed in the GWAS. Heatmap and contour plot for logistic regression-based liability scaled LDSC $h^2$ (**b**) and effect size ratio $rb$ (**c**) as functions of $Var(\tau E)$ and $Var(X\eta \circ E)$. Optimal values for $Var(\tau E)$ and $Var(X\eta \circ E)$, that is, for $\tau$ and $\eta$, respectively, comply with female prevalence, female heritability, and observed effect size ratio as well. The optimal $\tau$ turns out to be close to zero so that the environmental factor acts mostly via genetic interaction.

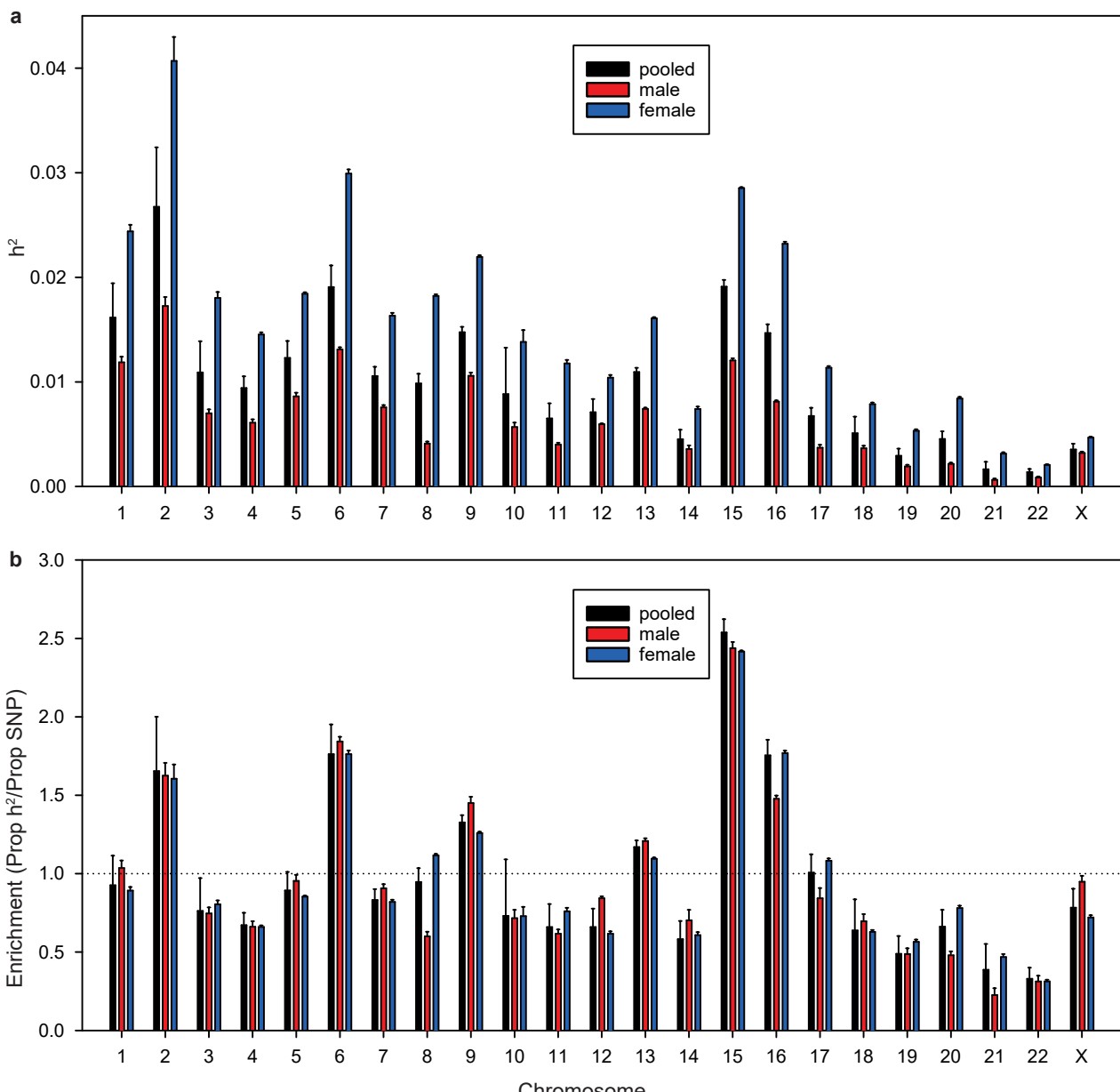

**Extended Data Fig. 5 | Per chromosome heritability estimation based on the EU-RLS-GENE dataset.** Heritability estimates for each chromosome. **a**, Overall heritability of SNPs on each chromosome. The height of the bar represents the point estimate of the heritability, and the error bars indicate the standard error of this point estimate. **b**, Enrichment of heritability, which is defined as the proportion of SNP-heritability divided by the proportion of SNPs in each chromosome. The height of the bar represents the point estimate of the enrichment of heritability, and the error bars indicate the standard error of this point estimate.

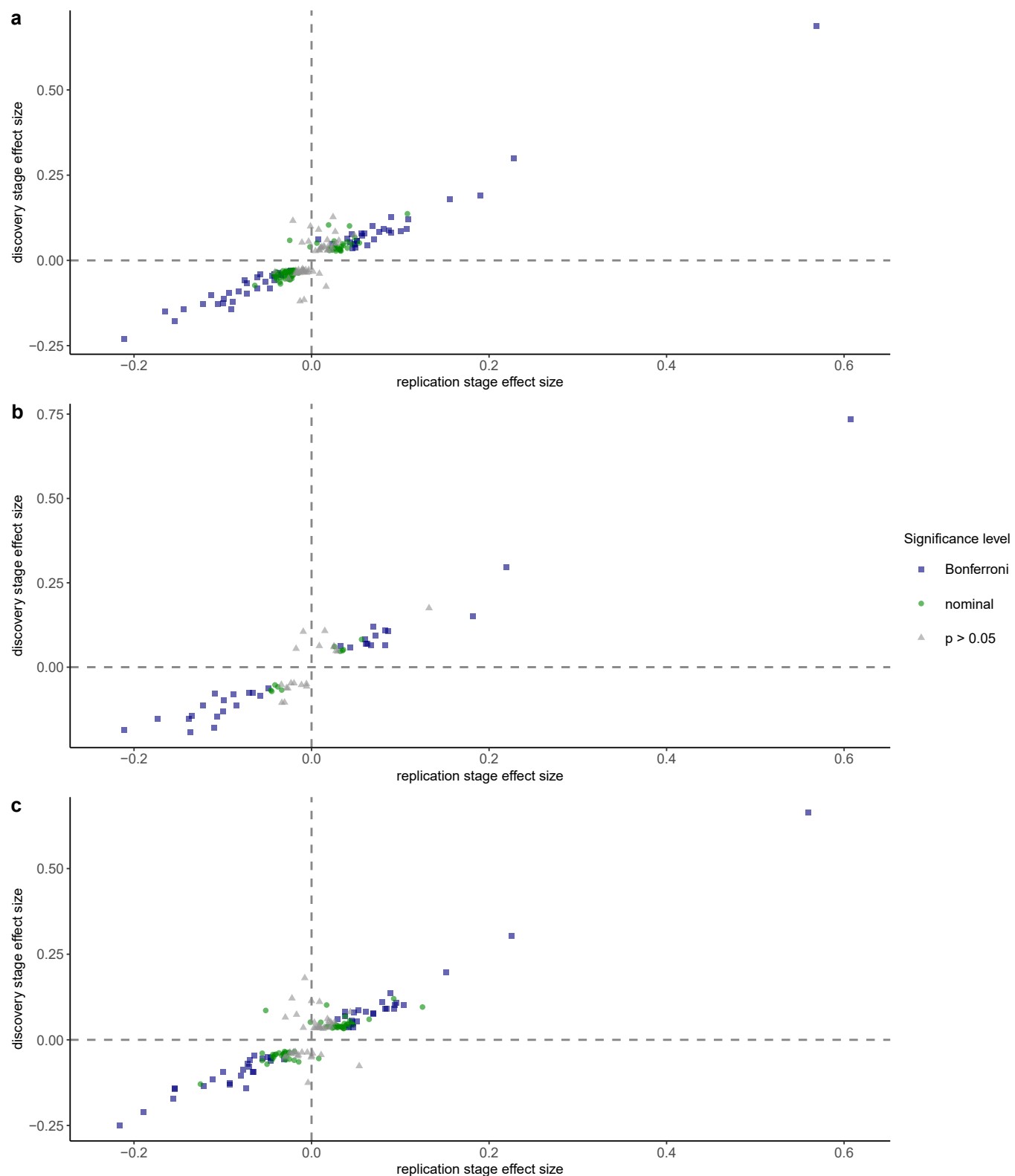

**Extended Data Fig. 6 | Replication of lead SNPs in independent validation samples.** Association results of replication stage. The effect size (beta) of the replication analysis is plotted against the effect size (beta) of the discovery stage for genome-wide significant lead SNPs. The color-coding and symbol shape indicate the strength of the association signal in the replication stage meta-analysis (nominal two-sided P value of random-effects meta-analysis).

Blue square, Bonferroni-corrected significance; green circle, nominal significance; grey triangle, not significant. **a**, Pooled meta-analysis with Bonferroni threshold set at 0.000255, correcting for 196 lead SNPs. **b**, Male-specific meta-analysis with Bonferroni threshold set at 0.00082, correcting for 61 lead SNPs. **c**, Female-specific meta-analysis with Bonferroni threshold set at 0.000318, correcting for 157 lead SNPs.

**Extended Data Table 1 | Lead SNPs with significant heterogeneity of effect sizes between sexes**

| Locus | Lead SNP | $P_{het}$ | $Beta_{male}$ (se) | P-value$_{male}$ | $Beta_{female}$ (se) | P-value$_{female}$ |
|---|---|---|---|---|---|---|
| 2:189979936-190692798 | rs12693542 | 0.0002 | -0.015 (0.009) | 0.0873 | -0.056 (0.007) | $1.1 \times 10^{-17}$ |
| 3:130247739-131344457 | rs9859139 | $8.6 \times 10^{-5}$ | 0.06 (0.008) | $4.9 \times 10^{-13}$ | 0.1 (0.006) | $3.0 \times 10^{-60}$ |
| 3:169636938-170166274 | rs9856511 | $9.6 \times 10^{-5}$ | -0.027 (0.008) | $3.1 \times 10^{-8}$ | -0.010 (0.008) | 0.202 |
| 14:21616150-21629436 | rs10498276 | $2.6 \times 10^{-5}$ | 0.054 (0.01) | $9.9 \times 10^{-8}$ | 0.111 (0.009) | $1.3 \times 10^{-36}$ |
| 15:67253093-68711447 | rs868036 | $3.4 \times 10^{-6}$ | 0.145 (0.009) | $2.0 \times 10^{-56}$ | 0.196 (0.006) | $6.7 \times 10^{-211}$ |
| 16:51886056-53340523 | rs3104769 | $1.7 \times 10^{-8}$ | -0.151 (0.009) | $9.9 \times 10^{-71}$ | -0.21 (0.006) | $9.1 \times 10^{-275}$ |

Locus, chromosomal position of risk locus (chr:start-end, hg19); lead SNP, N-GWAMA lead SNP for locus from pooled meta-analysis; $P_{het}$, nominal one-sided P value of Cochran's Q test for heterogeneity; $Beta_{male}$ and *P-value*$_{male}$, effect size estimate and nominal two-sided P value of male-only meta-analysis; $Beta_{female}$ and *P-value*$_{female}$, effect size estimate and nominal two-sided P value of female-only meta-analysis.

# Reporting Summary

## Statistics

For all statistical analyses, confirm that the following items are present in the figure legend, table legend, main text, or Methods section.

| n/a | Confirmed | |
|---|---|---|
| ☐ | ☒ | The exact sample size (*n*) for each experimental group/condition, given as a discrete number and unit of measurement |
| ☒ | ☐ | A statement on whether measurements were taken from distinct samples or whether the same sample was measured repeatedly |
| ☐ | ☒ | The statistical test(s) used AND whether they are one- or two-sided *Only common tests should be described solely by name; describe more complex techniques in the Methods section.* |
| ☐ | ☒ | A description of all covariates tested |
| ☐ | ☒ | A description of any assumptions or corrections, such as tests of normality and adjustment for multiple comparisons |
| ☐ | ☒ | A full description of the statistical parameters including central tendency (e.g. means) or other basic estimates (e.g. regression coefficient) AND variation (e.g. standard deviation) or associated estimates of uncertainty (e.g. confidence intervals) |
| ☐ | ☒ | For null hypothesis testing, the test statistic (e.g. *F*, *t*, *r*) with confidence intervals, effect sizes, degrees of freedom and *P* value noted *Give P values as exact values whenever suitable.* |
| ☒ | ☐ | For Bayesian analysis, information on the choice of priors and Markov chain Monte Carlo settings |
| ☒ | ☐ | For hierarchical and complex designs, identification of the appropriate level for tests and full reporting of outcomes |
| ☐ | ☒ | Estimates of effect sizes (e.g. Cohen's *d*, Pearson's *r*), indicating how they were calculated |

*Our web collection on statistics for biologists contains articles on many of the points above.*

## Software and code

Policy information about availability of computer code

| | |
|---|---|
| Data collection | Illumina GenomeStudio 2.0 |
| Data analysis | We provide information on publicly available software and settings in the Online Methods and the Supplementary Note. For custom data analysis, we describe the theoretical background as well as the models used in detail in the Supplementary Note. Custom code scripts are available on Zenodo (https://doi.org/10.5281/zenodo.10804907). Publicly available software used in this study: PLINK (v1.90b6.7): https://www.cog-genomics.org/plink/1.9/ SNPTEST (v2.5.4)  https://www.chg.ox.ac.uk/~gav/snptest/ SAIGE (0.35.8.8): https://github.com/saigegit/SAIGE N-GWAMA (v1.2.6) https://github.com/baselmans/multivariate_GWAMA; METAL (release 2011-03-25): https://csg.sph.umich.edu/abecasis/metal/index.html METASOFT (v2.0.1): https://web.cs.ucla.edu/~eeskin/ GCTA (v1.93.0beta): https://yanglab.westlake.edu.cn/software/gcta/#Overview LDSC (v1.0.1): https://github.com/bulik/ldsc LDAK (v5.0): https://dougspeed.com/ldak/ LHC-MR (v0.0.0.9000): https://github.com/LizaDarrous/lhcMR DEPICT (v1 rel194): https://github.com/perslab/depict FUMA (v1.3.6a): https://fuma.ctglab.nl/ MAGMA (v1.08) https://cncr.nl/research/magma/ MetaXcan (v0.7.4) https://github.com/hakyimlab/MetaXcan eCAVIAR (v2.2): https://github.com/fhormoz/caviar |

CAVIARBF (v0.2.1): https://bitbucket.org/Wenan/caviarbf/src/master/
CELLECT (v1.3.0) and CELLEX (v1.2.1): https://github.com/perslab/CELLECT
MAGMA_celltyping (v2.0.0): https://github.com/neurogenomics/MAGMA_Celltyping
pycox (v0.2.1): https://github.com/havakv/pycox
PyTorch (v1.6.0): https://github.com/pytorch/pytorch
H2O autoML (v3.36.0.2): https://docs.h2o.ai/h2o/latest-stable/h2o-docs/automl.html
Sanger imputation server: https://imputation.sanger.ac.uk/
EAGLE2 (v2.0.5): https://alkesgroup.broadinstitute.org/Eagle/
PBWT (v3.1): https://github.com/richarddurbin/pbwt
Minimac3: https://genome.sph.umich.edu/wiki/Minimac3
R (v4.0.4 and v4.0.2): https://cran.r-project.org/
Rrvgo package (v1.2.0):https://bioconductor.org/packages/release/bioc/html/rrvgo.html
WGCNA package (v1.69): https://cran.r-project.org/web/packages/WGCNA/index.html
TwoSampleMR package (v0.5.6): https://mrcieu.github.io/TwoSampleMR/index.html
Coloc package (v5.1.0) https://chr1swallace.github.io/coloc/
BayesianTools package (v0.0.10): https://github.com/florianhartig/BayesianTools
Bigsnpr package (v1.12.2), implements LDpred2: https://privefl.github.io/bigsnpr/
gwasvcf package (v0.1.0): https://github.com/MRCIEU/gwasvcf
randomForestSRC package (v3.0.1): https://www.randomforestsrc.org/

For manuscripts utilizing custom algorithms or software that are central to the research but not yet described in published literature, software must be made available to editors and reviewers. We strongly encourage code deposition in a community repository (e.g. GitHub). See the Nature Portfolio guidelines for submitting code & software for further information.

# Data

Policy information about availability of data

All manuscripts must include a data availability statement. This statement should provide the following information, where applicable:

- Accession codes, unique identifiers, or web links for publicly available datasets
- A description of any restrictions on data availability
- For clinical datasets or third party data, please ensure that the statement adheres to our policy

Summary statistics of the meta-analysis are publicly available for the top 10,000 SNPs at Zenodo (https://doi.org/10.5281/zenodo.10804907). Summary statistics of the discovery stage International EU-RLS-GENE consortium GWAS and the INTERVAL GWAS are available at GWAS Catalog (https://www.ebi.ac.uk/gwas/) under accession codes GCST90399568, GCST90399569, GCST90399570, GCST90399571, GCST90399572, and GCST90399573.The full GWAS summary statistics for the 23andMe discovery data set will be made available through 23andMe to qualified researchers under an agreement with 23andMe that protects the privacy of the 23andMe participants. Datasets will be made available at no cost for academic use. Please visit https://research.23andme.com/collaborate/#dataset-access/ for more information and to apply to access the data.
Additional data used for tissue and cell-type enrichment analysis are available here: developmental (http://mousebrain.org/development/downloads.html) and adult single cell RNAseq datasets (http://mousebrain.org/adult/downloads.html) from the Mouse Brain Atlas (http://mousebrain.org/), the Human Gene Expression During Development dataset from the BBI-Allen Single Cell atlases (https://descartes.brotmanbaty.org/), the BrainSpan Developmental Transcriptome RNA-Seq dataset from the BrainSpan Atlas of the Developing Human Brain (https://www.brainspan.org/static/home), the V8 RNA-Seq dataset (GTEx_Analysis_2017-06-05_v8_RNASeQCv1.1.9_gene_reads.gct.gz) from GTEx (https://gtexportal.org/home/datasets), and the human C8 collection from MSigDb v7.4 (http://software.broadinstitute.org/gsea/msigdb/, with the legacy versions available at https://www.gsea-msigdb.org/gsea/downloads_archive.jsp after creating a user account with GSEA/MSigDB).
Summary statistics of GWAS for genetic correlation and MR analyses are available at the University of Bristol Integrative Epidemiology Unit OpenGWAS server https://gwas.mrcieu.ac.uk) and GWAS atlas (https://atlas.ctglab.nl/). Additional GWAS summary statistics for iron-related traits are available at https://www.fmrib.ox.ac.uk/ukbiobank/gwas_resources/index.html, https://open.win.ox.ac.uk/ukbiobank/big40/BIGv2/, and https://www.decode.com/summarydata/.
A complete list of sources used for annotation with FUMA is available at https://fuma.ctglab.nl/links and https://fuma.ctglab.nl/tutorial. Auxiliary files for use with MAGMA are available at https://ctg.cncr.nl/software/magma. Additional files for use with LDSC and LDAK are available at https://alkesgroup.broadinstitute.org/LDSCORE/.
Information about drug targets is available at the free-to-access database DrugBank Online (https://go.drugbank.com/).

# Human research participants

Policy information about studies involving human research participants and Sex and Gender in Research.

| | |
|---|---|
| Reporting on sex and gender | The study included both female and male individuals. A pooled analysis of all individuals was performed as well as analyses stratified by sex. Sex was determined by self-reporting and by genotyping. During genotyping data quality control, individuals with non-matching genotype-determined sex and self-reported sex were excluded and were not included in any further analyses. Sample sizes for the sex-specific discovery analyses were 78,333 cases and 844,872 controls in women and 38,314 cases and 701,594 controls in men. |
| Population characteristics | The discovery dataset for the pooled analysis included 116,647 cases and 1,546,466 controls. 78,333 cases and 844,872 controls were women and 38,314 cases and 701,594 controls were men.<br>All individuals were of European ancestry, determined by running PCA or MDS analysis on the genetic data. Individuals of non-European ancestry were excluded to avoid spurious associations due to population stratification.<br>Age and sex were used as covariates. A total of 9,196,648 common variants with minor allele frequency (MAF) ≥ 1% were available for statistical analysis in the discovery stage data.<br>The case phenotype "restless legs syndrome" (RLS) was determined by either clinical face-to-face interviews or using validated questionnaires for RLS cases, implementing the IRLSSG diagnostic criteria for RLS. In the 23andMe dataset, RLS |

| Recruitment | cases were defined by asking a single question to customers about having received an RLS diagnosis or therapy. |
|---|---|
| Recruitment | Participants were recruited<br>1) in a clinical setting: Cases were recruited in specialized outpatient clinics for movement disorders and in sleep clinics by conducting face-to-face diagnostic interviews to assess the IRLSSG diagnostic criteria for RLS<br>2) in cohorts of whole blood donors by self-report based on validated questionnaires for RLS (Cambridge-Hopkins Restless Legs Questionnaire)<br>3) in a direct-to-consumer genetic testing company customer database by self-report based on survey questions which assessed whether someone has ever been diagnosed with RLS or has ever received treatment for RLS.<br>Genetic correlations between the three GWAS with different case recruitment strategies were strong but indicated some degree of heterogeneity, therefore, a multivariate genome-wide-association meta-analysis approach was used (N-GWAMA). |
| Ethics oversight | All studies were approved by the respective local ethical committees and all participants provided informed consent. The EU-RLS-GENE study was approved by an institutional review board at the University Hospital of the Technical University Munich (2488/09). The INTERVAL dataset was approved by the National Research Ethics Service Committee East of England - Cambridge East (REC: 11/EE/0538). 23andMe Participants provided informed consent under a protocol approved by the external AAHRPP-accredited IRB, Ethical & Independent (E&I) Review Services. As of 2022, E&I Review Services is part of Salus IRB (https://www.versiticlinicaltrials.org/salusirb). The deCODE dataset was approved by the National Bioethics Committee of Iceland. The DBDS dataset was approved by The Scientific Ethical Committee of Central Denmark (M-20090237) and by the Danish Data Protection agency (30-0444). GWAS studies in DBDS were approved by the National Ethical Committee (NVK-1700407). The Emory dataset was approved by an institutional review board at Emory University, Atlanta, Georgia, US (HIC ID 133-98). |

Note that full information on the approval of the study protocol must also be provided in the manuscript.

# Field-specific reporting

Please select the one below that is the best fit for your research. If you are not sure, read the appropriate sections before making your selection.

☒ Life sciences ☐ Behavioural & social sciences ☐ Ecological, evolutionary & environmental sciences

For a reference copy of the document with all sections, see nature.com/documents/nr-reporting-summary-flat.pdf

# Life sciences study design

All studies must disclose on these points even when the disclosure is negative.

| Sample size | No sample-size calculation was performed. Restless legs syndrome is a polygenic trait for which heritability is not yet fully explained. Therefore, we collected all available (at the time of initiating the study) GWAS datasets for this phenotype into a discovery dataset. This dataset is about 8 times larger than the datasets used in previous GWAS on RLS, therefore could be expected to provide a reasonable increase in study power. |
|---|---|
| Data exclusions | Phenotypic data: only RLS patients fullfilling the diagnostic criteria based on either a face-to-face interview, validated questionnaires, or a single question about RLS diagnosis/treatment were included in the study.<br>Genetic data: Standard SNP and sample GWAS quality control procedures were applied to exclude low quality data. |
| Replication | Data for all independent lead SNPs of the discovery stage was obtained in independent replication dataset consisting of 29,028 RLS cases and 398,815 controls. 71% of the lead SNPs from the pooled discovery meta-analysis were nominally significant in the replication (p < 0.05) and there was a high positive correlation between the effect size estimates of the discovery stage and the replication dataset. A joint analysis of discovery and replication datasets revealed that all lead SNPs of the discovery pooled and sex-specific meta-analyses reached Bonferroni-corrected significance. |
| Randomization | GWAS are observational genetic studies and not randomized experiments. For a GWAS, individuals are assigned to either the case group (indiviuals affected by RLS) or the control group (unaffected individuals). |
| Blinding | Meta-analysis of GWAS and functional GWAS interpretation do not require blinding because GWAS are observational genetic studies and not randomized experiments. |

# Behavioural & social sciences study design

All studies must disclose on these points even when the disclosure is negative.

| Study description | *Briefly describe the study type including whether data are quantitative, qualitative, or mixed-methods (e.g. qualitative cross-sectional,* |
|---|---|

| | |
|---|---|
| Study description | *quantitative experimental, mixed-methods case study).* |
| Research sample | *State the research sample (e.g. Harvard university undergraduates, villagers in rural India) and provide relevant demographic information (e.g. age, sex) and indicate whether the sample is representative. Provide a rationale for the study sample chosen. For studies involving existing datasets, please describe the dataset and source.* |
| Sampling strategy | *Describe the sampling procedure (e.g. random, snowball, stratified, convenience). Describe the statistical methods that were used to predetermine sample size OR if no sample-size calculation was performed, describe how sample sizes were chosen and provide a rationale for why these sample sizes are sufficient. For qualitative data, please indicate whether data saturation was considered, and what criteria were used to decide that no further sampling was needed.* |
| Data collection | *Provide details about the data collection procedure, including the instruments or devices used to record the data (e.g. pen and paper, computer, eye tracker, video or audio equipment) whether anyone was present besides the participant(s) and the researcher, and whether the researcher was blind to experimental condition and/or the study hypothesis during data collection.* |
| Timing | *Indicate the start and stop dates of data collection. If there is a gap between collection periods, state the dates for each sample cohort.* |
| Data exclusions | *If no data were excluded from the analyses, state so OR if data were excluded, provide the exact number of exclusions and the rationale behind them, indicating whether exclusion criteria were pre-established.* |
| Non-participation | *State how many participants dropped out/declined participation and the reason(s) given OR provide response rate OR state that no participants dropped out/declined participation.* |
| Randomization | *If participants were not allocated into experimental groups, state so OR describe how participants were allocated to groups, and if allocation was not random, describe how covariates were controlled.* |

# Ecological, evolutionary & environmental sciences study design

All studies must disclose on these points even when the disclosure is negative.

| | |
|---|---|
| Study description | *Briefly describe the study. For quantitative data include treatment factors and interactions, design structure (e.g. factorial, nested, hierarchical), nature and number of experimental units and replicates.* |
| Research sample | *Describe the research sample (e.g. a group of tagged Passer domesticus, all Stenocereus thurberi within Organ Pipe Cactus National Monument), and provide a rationale for the sample choice. When relevant, describe the organism taxa, source, sex, age range and any manipulations. State what population the sample is meant to represent when applicable. For studies involving existing datasets, describe the data and its source.* |
| Sampling strategy | *Note the sampling procedure. Describe the statistical methods that were used to predetermine sample size OR if no sample-size calculation was performed, describe how sample sizes were chosen and provide a rationale for why these sample sizes are sufficient.* |
| Data collection | *Describe the data collection procedure, including who recorded the data and how.* |
| Timing and spatial scale | *Indicate the start and stop dates of data collection, noting the frequency and periodicity of sampling and providing a rationale for these choices. If there is a gap between collection periods, state the dates for each sample cohort. Specify the spatial scale from which the data are taken* |
| Data exclusions | *If no data were excluded from the analyses, state so OR if data were excluded, describe the exclusions and the rationale behind them, indicating whether exclusion criteria were pre-established.* |
| Reproducibility | *Describe the measures taken to verify the reproducibility of experimental findings. For each experiment, note whether any attempts to repeat the experiment failed OR state that all attempts to repeat the experiment were successful.* |
| Randomization | *Describe how samples/organisms/participants were allocated into groups. If allocation was not random, describe how covariates were controlled. If this is not relevant to your study, explain why.* |
| Blinding | *Describe the extent of blinding used during data acquisition and analysis. If blinding was not possible, describe why OR explain why blinding was not relevant to your study.* |

Did the study involve field work?  ☐ Yes   ☐ No

## Field work, collection and transport

| | |
|---|---|
| Field conditions | *Describe the study conditions for field work, providing relevant parameters (e.g. temperature, rainfall).* |
| Location | *State the location of the sampling or experiment, providing relevant parameters (e.g. latitude and longitude, elevation, water depth).* |

| Access & import/export | *Describe the efforts you have made to access habitats and to collect and import/export your samples in a responsible manner and in compliance with local, national and international laws, noting any permits that were obtained (give the name of the issuing authority, the date of issue, and any identifying information).* |
|---|---|
| Disturbance | *Describe any disturbance caused by the study and how it was minimized.* |

# Reporting for specific materials, systems and methods

We require information from authors about some types of materials, experimental systems and methods used in many studies. Here, indicate whether each material, system or method listed is relevant to your study. If you are not sure if a list item applies to your research, read the appropriate section before selecting a response.

### Materials & experimental systems

| n/a | Involved in the study |
|---|---|
| ☒ | ☐ Antibodies |
| ☒ | ☐ Eukaryotic cell lines |
| ☒ | ☐ Palaeontology and archaeology |
| ☒ | ☐ Animals and other organisms |
| ☒ | ☐ Clinical data |
| ☒ | ☐ Dual use research of concern |

### Methods

| n/a | Involved in the study |
|---|---|
| ☒ | ☐ ChIP-seq |
| ☒ | ☐ Flow cytometry |
| ☒ | ☐ MRI-based neuroimaging |

## Antibodies

| Antibodies used | *Describe all antibodies used in the study; as applicable, provide supplier name, catalog number, clone name, and lot number.* |
|---|---|
| Validation | *Describe the validation of each primary antibody for the species and application, noting any validation statements on the manufacturer's website, relevant citations, antibody profiles in online databases, or data provided in the manuscript.* |

## Eukaryotic cell lines

Policy information about cell lines and Sex and Gender in Research

| Cell line source(s) | *State the source of each cell line used and the sex of all primary cell lines and cells derived from human participants or vertebrate models.* |
|---|---|
| Authentication | *Describe the authentication procedures for each cell line used OR declare that none of the cell lines used were authenticated.* |
| Mycoplasma contamination | *Confirm that all cell lines tested negative for mycoplasma contamination OR describe the results of the testing for mycoplasma contamination OR declare that the cell lines were not tested for mycoplasma contamination.* |
| Commonly misidentified lines (See ICLAC register) | *Name any commonly misidentified cell lines used in the study and provide a rationale for their use.* |

## Palaeontology and Archaeology

| Specimen provenance | *Provide provenance information for specimens and describe permits that were obtained for the work (including the name of the issuing authority, the date of issue, and any identifying information). Permits should encompass collection and, where applicable, export.* |
|---|---|
| Specimen deposition | *Indicate where the specimens have been deposited to permit free access by other researchers.* |
| Dating methods | *If new dates are provided, describe how they were obtained (e.g. collection, storage, sample pretreatment and measurement), where they were obtained (i.e. lab name), the calibration program and the protocol for quality assurance OR state that no new dates are provided.* |

☐ Tick this box to confirm that the raw and calibrated dates are available in the paper or in Supplementary Information.

| Ethics oversight | *Identify the organization(s) that approved or provided guidance on the study protocol, OR state that no ethical approval or guidance was required and explain why not.* |
|---|---|

Note that full information on the approval of the study protocol must also be provided in the manuscript.

# Animals and other research organisms

Policy information about studies involving animals; ARRIVE guidelines recommended for reporting animal research, and Sex and Gender in Research

| | |
|---|---|
| Laboratory animals | *For laboratory animals, report species, strain and age OR state that the study did not involve laboratory animals.* |
| Wild animals | *Provide details on animals observed in or captured in the field; report species and age where possible. Describe how animals were caught and transported and what happened to captive animals after the study (if killed, explain why and describe method; if released, say where and when) OR state that the study did not involve wild animals.* |
| Reporting on sex | *Indicate if findings apply to only one sex; describe whether sex was considered in study design, methods used for assigning sex. Provide data disaggregated for sex where this information has been collected in the source data as appropriate; provide overall numbers in this Reporting Summary. Please state if this information has not been collected. Report sex-based analyses where performed, justify reasons for lack of sex-based analysis.* |
| Field-collected samples | *For laboratory work with field-collected samples, describe all relevant parameters such as housing, maintenance, temperature, photoperiod and end-of-experiment protocol OR state that the study did not involve samples collected from the field.* |
| Ethics oversight | *Identify the organization(s) that approved or provided guidance on the study protocol, OR state that no ethical approval or guidance was required and explain why not.* |

Note that full information on the approval of the study protocol must also be provided in the manuscript.

# Clinical data

Policy information about clinical studies
All manuscripts should comply with the ICMJE guidelines for publication of clinical research and a completed CONSORT checklist must be included with all submissions.

| | |
|---|---|
| Clinical trial registration | *Provide the trial registration number from ClinicalTrials.gov or an equivalent agency.* |
| Study protocol | *Note where the full trial protocol can be accessed OR if not available, explain why.* |
| Data collection | *Describe the settings and locales of data collection, noting the time periods of recruitment and data collection.* |
| Outcomes | *Describe how you pre-defined primary and secondary outcome measures and how you assessed these measures.* |

# Dual use research of concern

Policy information about dual use research of concern

## Hazards

Could the accidental, deliberate or reckless misuse of agents or technologies generated in the work, or the application of information presented in the manuscript, pose a threat to:

No | Yes
☐ | ☐ Public health
☐ | ☐ National security
☐ | ☐ Crops and/or livestock
☐ | ☐ Ecosystems
☐ | ☐ Any other significant area

## Experiments of concern

Does the work involve any of these experiments of concern:

| No | Yes | |
|----|-----|--|
| ☐ | ☐ | Demonstrate how to render a vaccine ineffective |
| ☐ | ☐ | Confer resistance to therapeutically useful antibiotics or antiviral agents |
| ☐ | ☐ | Enhance the virulence of a pathogen or render a nonpathogen virulent |
| ☐ | ☐ | Increase transmissibility of a pathogen |
| ☐ | ☐ | Alter the host range of a pathogen |
| ☐ | ☐ | Enable evasion of diagnostic/detection modalities |
| ☐ | ☐ | Enable the weaponization of a biological agent or toxin |
| ☐ | ☐ | Any other potentially harmful combination of experiments and agents |

# ChIP-seq

## Data deposition

☐ Confirm that both raw and final processed data have been deposited in a public database such as GEO.

☐ Confirm that you have deposited or provided access to graph files (e.g. BED files) for the called peaks.

| | |
|--|--|
| Data access links<br>*May remain private before publication.* | *For "Initial submission" or "Revised version" documents, provide reviewer access links. For your "Final submission" document, provide a link to the deposited data.* |
| Files in database submission | *Provide a list of all files available in the database submission.* |
| Genome browser session<br>(e.g. UCSC) | *Provide a link to an anonymized genome browser session for "Initial submission" and "Revised version" documents only, to enable peer review. Write "no longer applicable" for "Final submission" documents.* |

## Methodology

| | |
|--|--|
| Replicates | *Describe the experimental replicates, specifying number, type and replicate agreement.* |
| Sequencing depth | *Describe the sequencing depth for each experiment, providing the total number of reads, uniquely mapped reads, length of reads and whether they were paired- or single-end.* |
| Antibodies | *Describe the antibodies used for the ChIP-seq experiments; as applicable, provide supplier name, catalog number, clone name, and lot number.* |
| Peak calling parameters | *Specify the command line program and parameters used for read mapping and peak calling, including the ChIP, control and index files used.* |
| Data quality | *Describe the methods used to ensure data quality in full detail, including how many peaks are at FDR 5% and above 5-fold enrichment.* |
| Software | *Describe the software used to collect and analyze the ChIP-seq data. For custom code that has been deposited into a community repository, provide accession details.* |

# Flow Cytometry

## Plots

Confirm that:

☐ The axis labels state the marker and fluorochrome used (e.g. CD4-FITC).

☐ The axis scales are clearly visible. Include numbers along axes only for bottom left plot of group (a 'group' is an analysis of identical markers).

☐ All plots are contour plots with outliers or pseudocolor plots.

☐ A numerical value for number of cells or percentage (with statistics) is provided.

## Methodology

| | |
|--|--|
| Sample preparation | *Describe the sample preparation, detailing the biological source of the cells and any tissue processing steps used.* |
| Instrument | *Identify the instrument used for data collection, specifying make and model number.* |

| Software | *Describe the software used to collect and analyze the flow cytometry data. For custom code that has been deposited into a community repository, provide accession details.* |
|---|---|
| Cell population abundance | *Describe the abundance of the relevant cell populations within post-sort fractions, providing details on the purity of the samples and how it was determined.* |
| Gating strategy | *Describe the gating strategy used for all relevant experiments, specifying the preliminary FSC/SSC gates of the starting cell population, indicating where boundaries between "positive" and "negative" staining cell populations are defined.* |

☐ Tick this box to confirm that a figure exemplifying the gating strategy is provided in the Supplementary Information.

# Magnetic resonance imaging

## Experimental design

| Design type | *Indicate task or resting state; event-related or block design.* |
|---|---|
| Design specifications | *Specify the number of blocks, trials or experimental units per session and/or subject, and specify the length of each trial or block (if trials are blocked) and interval between trials.* |
| Behavioral performance measures | *State number and/or type of variables recorded (e.g. correct button press, response time) and what statistics were used to establish that the subjects were performing the task as expected (e.g. mean, range, and/or standard deviation across subjects).* |

## Acquisition

| Imaging type(s) | *Specify: functional, structural, diffusion, perfusion.* |
|---|---|
| Field strength | *Specify in Tesla* |
| Sequence & imaging parameters | *Specify the pulse sequence type (gradient echo, spin echo, etc.), imaging type (EPI, spiral, etc.), field of view, matrix size, slice thickness, orientation and TE/TR/flip angle.* |
| Area of acquisition | *State whether a whole brain scan was used OR define the area of acquisition, describing how the region was determined.* |

Diffusion MRI ☐ Used ☐ Not used

## Preprocessing

| Preprocessing software | *Provide detail on software version and revision number and on specific parameters (model/functions, brain extraction, segmentation, smoothing kernel size, etc.).* |
|---|---|
| Normalization | *If data were normalized/standardized, describe the approach(es): specify linear or non-linear and define image types used for transformation OR indicate that data were not normalized and explain rationale for lack of normalization.* |
| Normalization template | *Describe the template used for normalization/transformation, specifying subject space or group standardized space (e.g. original Talairach, MNI305, ICBM152) OR indicate that the data were not normalized.* |
| Noise and artifact removal | *Describe your procedure(s) for artifact and structured noise removal, specifying motion parameters, tissue signals and physiological signals (heart rate, respiration).* |
| Volume censoring | *Define your software and/or method and criteria for volume censoring, and state the extent of such censoring.* |

## Statistical modeling & inference

| Model type and settings | *Specify type (mass univariate, multivariate, RSA, predictive, etc.) and describe essential details of the model at the first and second levels (e.g. fixed, random or mixed effects; drift or auto-correlation).* |
|---|---|
| Effect(s) tested | *Define precise effect in terms of the task or stimulus conditions instead of psychological concepts and indicate whether ANOVA or factorial designs were used.* |

Specify type of analysis: ☐ Whole brain ☐ ROI-based ☐ Both

| Statistic type for inference (See Eklund et al. 2016) | *Specify voxel-wise or cluster-wise and report all relevant parameters for cluster-wise methods.* |
|---|---|
| Correction | *Describe the type of correction and how it is obtained for multiple comparisons (e.g. FWE, FDR, permutation or Monte Carlo).* |

## Models & analysis

nature portfolio | reporting summary

| n/a | Involved in the study |
|-----|------------------------|
| ☐ | ☐ Functional and/or effective connectivity |
| ☐ | ☐ Graph analysis |
| ☐ | ☐ Multivariate modeling or predictive analysis |

Functional and/or effective connectivity

*Report the measures of dependence used and the model details (e.g. Pearson correlation, partial correlation, mutual information).*

Graph analysis

*Report the dependent variable and connectivity measure, specifying weighted graph or binarized graph, subject- or group-level, and the global and/or node summaries used (e.g. clustering coefficient, efficiency, etc.).*

Multivariate modeling and predictive analysis

*Specify independent variables, features extraction and dimension reduction, model, training and evaluation metrics.*

March 2021

