## [Peer Review File · Nature Genetics]

Peer Review Information

Manuscript Title: Genome-wide meta-analyses of restless legs syndrome yield insights into genetic architecture, disease biology, and risk prediction

Corresponding author name(s): Dr Barbara Schormair

Editorial Notes:

Transferred manuscripts This document only contains reviewer comments, rebuttal and decision letters for versions considered at Nature Genetics.

Reviewer Comments & Decisions:

Decision Letter, initial version:

24th May 2023

Dear Barbara,

Your Article "GWAS meta-analyses of restless legs syndrome identify 164 risk loci, highlight sex-specific effects, and advance risk prediction and treatment" has now been seen by two referees. You will see from their comments below that, while they find your work of interest, they have raised several relevant points. We are interested in the possibility of publishing your study in Nature Genetics, but we would like to consider your response to these points in the form of a revised manuscript before we make a final decision on publication.

To guide the scope of the revisions, the editors discuss the referee reports in detail within the team, including with the chief editor, with a view to identifying key priorities that should be addressed in revision, and sometimes overruling referee requests that are deemed beyond the scope of the current study. In this case, we ask that you address all technical queries related to the association analyses and their interpretation, extending the analyses where feasible and revising the presentation for clarity as requested by the referees. We hope you will find this prioritized set of referee points to be useful when revising your study. Please do not hesitate to get in touch if you would like to discuss these issues further.

We therefore invite you to revise your manuscript taking into account all reviewer and editor comments. Please highlight all changes in the manuscript text file. At this stage, we will need you to upload a copy of the manuscript in MS Word .docx or similar editable format.

*2) If you have not done so already, please begin to revise your manuscript so that it conforms to our Article format instructions, available here.

*3) Include a revised version of any required Reporting Summary:

Please be aware of our guidelines on digital image standards.

[redacted]

We hope to receive your revised manuscript within 8-12 weeks. If you cannot send it within this time, please let us know.

Sincerely,
Kyle

Kyle Vogan, PhD
Senior Editor
Nature Genetics
<https://orcid.org/0000-0001-9565-9665>

Referee expertise:

Referee #1: Genetics, sleep disorders

Referee #2: Genetics, sleep disorders

Reviewers' Comments:

Reviewer #1:
Remarks to the Author:

"GWAS meta-analyses of restless legs syndrome identify 164 risk loci, highlight sex-specific effects, and advance risk prediction and treatment"

Summary of key results:

Schormair et al. investigate common genetic variants associated with Restless Legs Syndrome in people of European ancestry. They identify 164 loci (196 lead SNPs) and assess heritability, sex-specific effects, tissue and pathway enrichment, druggable targets, genetic correlations, causal relationships, and disease prediction models.

Originality and significance:

Schormair and colleagues provide insight into new treatment and disease prediction avenues for RLS from the genetic findings marking an advance in the field.

Data and Methodology:

This investigation is incredibly thorough often utilizing multiple analytical methods to validate their findings and models. The methods are appropriate and utilize metrics and sensitivity analyses as needed to enable solid conclusions.

This manuscript is really well done, and I only have a few minor comments/thoughts.

1. In the intro (line 16-17), the authors mention how current PRS are limited and likely to underperform. It would also be helpful to add a brief sentence to the intro and discussion about extensions to other ancestry groups in an effort to reduce bias in prediction utility.

2. In the methods – some cohorts use linear models and others use mixed models for the genetic

analysis. What is the level of relatedness in each cohort and how did this choice impact the sample size?

3. I found Supplementary Table 2 ("Heritability estimates of pooled and sex-specific GWAS and meta-analyses") difficult to follow. Perhaps the addition of a table footnote explaining the different columns/methods would be helpful, as well as separating pooled from sex-specific into two tables.

4. In the prediction model, the results indicate a PRS x age interaction, with the PRS being more impactful at a younger age. Does this perhaps warrant an age-stratified GWAS?

5. For the sex-specific effects, there is an increased heritability in women. The authors explain this as an unobserved environmental risk factor. What would be the next steps to identify what this risk factor might be?

6. With tissue and pathway enrichment demonstrating developmental pathways, is there any thought to why RLS is a later onset disease? Might there be earlier subclinical impacts? Is early (e.g. infant) genetic screening warranted?

7. For the causal relationships, the bidirectional relationship between RLS/insomnia/depression is mentioned as a complicated relationship which may include undiagnosed RLS cases in these other phenotypes. Do the authors think GWAS by subtraction (eg. GenomicSEM) might reveal unique genetic components? Or are there other methods to distinguish the undiagnosed RLS cases?

8. Fig 1 is a little confusing and hard to read the underlying pathway names.

9. The colors used in Fig 5 make it hard to distinguish the lines.

10. The grayscale in Supp Fig 5 makes it hard to differentiate the three bars.

Reviewer #2:

Remarks to the Author:

Schormair et al present a genome-wide association study meta-analysis of 3 studies (EU-RLS-GENE consortium, INTERVAL study, 23andMe) for Restless Legs Syndrome (RLS), comprising of 116,403 cases and 1,473,166 controls of European ancestry. The authors report identifying 161 loci on the autosomes, with 131 novel and all 22 loci previously reported confirmed at $P < 5 \times 10^{-8}$. After conditional analysis, a total of 193 independent SNPs across the 161 loci were identified. In addition, sex-stratified analysis was also performed. The authors subsequently perform gene-set enrichment analysis to identify pathways implicated in RLS based on the GWAS summary stats. In addition, the authors use fine-mapping, genomic annotations, and co-localisation analyses to prioritise candidate causal genes. Analysis of genetic correlations were performed to prioritise traits for bi-directional MR analysis prior. Finally, the authors present analyses of the development of a genetic predictor using a linear models (with and without interaction terms), random forests and deep neural networks.

Overall, this does provide some advances in the understanding of the genetic architecture and pathways involved in RLS. However, I do have some comments for the authors:

Introduction:

"However, a substantial proportion of heritability in RLS remains 15 unaccounted for. The implications of this are that polygenic risk scores (PRS) for RLS underperform..."

OK but numerous studies have now shown this is unlikely to be fully resolved without inclusion of individuals of diverse ancestries – not Europeans. This is only mentioned right at the very end.

"close to 1.5 million controls"
Give actual number (1,473,166).

Can the authors expand on what they mean by "In order to provide opportunities for immediate translation into clinical use"? There have been few examples of this from GWAS. For example, a potential drug target is no sure guarantee of a compound being developed that passes all phases of a clinical trial. I also do not see how performing MR would lead into immediate translation either.

Results

- Pooled autosomal GWAS analysis:

Can the authors hypothesise why the genetic correlations were different among the 3 studies – for example, is this likely due to the heterogeneity in the phenotype definition, differences in analytical software/models applied, etc?

What is meant by $(\lambda)_{(subscript "1000")}$?

Have the authors tested the whether the differences in the heritability estimates between the three studies are statistically significant? For example, for the estimates given in the main text, I note that some of point estimates lie within 95%CI of some of the other estimates.

- Sex-stratified autosomal GWAS and meta-analyses

Please add "statistically" prior to the word "significant" (and throughout).

"but both other cohorts individually showed higher estimates for heritability in females compared to males ($p = 0.07$ in EU-RLS-GENE; $p = 0.09$ in 23andMe, Supplementary Table 2)." - Linked to comment above about presentation of ST2, can the authors confirm which values in ST2 these P-values are based on, and can the authors confirm what test has been performed to determine heterogeneity of heritability estimates?

"Comparing the two sex-specific meta-analyses showed a high genetic correlation of 0.96 ($se = 0.018$), however, the remaining small divergence was significant ($p = 0.044$), Supplementary Table 2"
– Can the authors confirm how this test was performed?

The last sentence starts "The most significant sex differences" – this is a bit counter intuitive if basing

on heterogeneity of effect estimates rather than being statistically significant in one sex and not in another – for which there are three examples out of the six. Also, the authors should note that the difference for the loci highlighted are small (~ 0.05 units) and the effects are directionally consistent.

While I appreciate the effort that has gone into the simulation work, is it not possible that the reason for the differences in heritability estimates are more simply due to the proportion of males and females that come from each of the three studies, where each study has defined RLS using different measures with some having higher heritability – over and above some unknown potential GxE interaction? The authors should provide a breakdown of the N for each sex from each of the three studies as well.

- X-chromosome analysis

It is not clear what the importance of links to neuroticism and estrogen mediated transcription is and why this is in the results and not discussion. Can the authors provide more narrative for links to RLS or place in discussion.

- Assessment of lead variants in independent cohorts

•“Assessment of lead variants in independent cohorts” These are not “independent cohorts” – they are out-of-sample and biases from each of the studies may carry through to these subsets. Please do not refer to these as “independent cohorts”.

- Functional annotation and biological mechanisms

“Excluding 12 SNPs not reaching genome-wide significance in the joint analysis of 1 discovery and validation did not change the results. (Extended Data Figure 6).” – This does not seem to be the correct figure.

- Genetic correlations and MR analysis

There are a number of places where subscripting should be clarified by the authors – for example, it is not clear what $a_{\text{RLS-diabetes2}} = 0.99$ refers to - is this supposed to be the causal effect estimate?

- Development and validation of a risk prediction model

How do these models compare with widely used models for PRS generation (LDPRED2, PRSICE, etc.)?

Other Comments:

Figure 1 needs to be much clearer – some text is impossible to read

Supplementary Tables

Supplementary Table 2 should be re-ordered to make references from the main text easier to navigate

when reading the table.

Supplementary Table 3:

Some of the cells contain #REF!

A p-value of 6.833632e-482 is presented. Please would the authors check this

Supplementary Table 4:

P-values of 2.17332e-747 and 3.428426e-447 are present. Please would the authors check these.

Extended Data Figures

The legend for Extended Data 3 starts "Extended data figure 2 legend"

Author Rebuttal to Initial comments

Reviewers' Comments:

Reviewer #1:

Remarks to the Author:

"GWAS meta-analyses of restless legs syndrome identify 164 risk loci, highlight sex-specific effects, and advance risk prediction and treatment"

Summary of key results:

Schormair et al. investigate common genetic variants associated with Restless Legs Syndrome in people of European ancestry. They identify 164 loci (196 lead SNPs) and assess heritability, sex-specific effects, tissue and pathway enrichment, druggable targets, genetic correlations, causal relationships, and disease prediction models.

Originality and significance:

Schormair and colleagues provide insight into new treatment and disease prediction avenues for RLS from the genetic findings marking an advance in the field.

Data and Methodology:

This investigation is incredibly thorough, often utilizing multiple analytical methods to validate their findings and models. The methods are appropriate and utilize metrics and sensitivity analyses as needed to enable solid conclusions.

This manuscript is really well done, and I only have a few minor comments/thoughts.

1. In the intro (line 16-17), the authors mention how current PRS are limited and likely to underperform. It would also be helpful to add a brief sentence to the intro and discussion about extensions to other ancestry groups in an effort to reduce bias in prediction utility.

AUTHOR RESPONSE: We fully agree that polygenic risk scores must be built and evaluated across diverse ancestries to reduce bias in their predictive utility. Unfortunately, there is a scarcity of genetic as well as epidemiologic studies focusing on RLS in populations of non-European ancestry. Therefore, we could not extend our analyses beyond European populations in the current study. Our introductory paragraph was intended to highlight the limitations that exist even within the well-studied European ancestry populations.

Considering your comment and the related comment #1 of reviewer 2, we have restructured the introduction and discussion. We tried to focus the introduction on topics that we could address in our current study with the data available to us, while covering study-specific limitations as well as general limitations of RLS research in the discussion. We do not intend to downplay the importance of including diverse ancestries, but we feel that the discussion section is the more suitable location to bring these limitations to the reader's attention.

In line with this view, we have adapted the Introduction and Discussion of the manuscript accordingly (please see the revised manuscript files).

2. In the methods – some cohorts use linear models and others use mixed models for the genetic analysis. What is the level of relatedness in each cohort and how did this choice impact the sample size?

AUTHOR RESPONSE: We would like to point out that the sample sizes reported in the paper are the sample sizes after quality control, i.e., after the exclusion of related individuals.

EU-RLS-GENE: There were about 7% related individuals (cutoff used: $PIHAT \geq 0.09375$, represents the halfway point between 3rd-degree and 4th-degree relatives). From each pair of related individuals, the sample with the lower genotyping quality was removed during quality control. Therefore, 3.5% of samples were removed due to relatedness.

INTERVAL: There were about 5% related individuals in the discovery cohort (cutoff used: $PIHAT \geq 0.1875$, representing the halfway point between 2nd-degree and 3rd-degree

relatives). INTERVAL had the smallest number of RLS cases of all three studies, therefore using SAIGE to keep related individuals in the dataset was more relevant for study power than in the two larger cohorts of EU-RLS-GENE and 23andMe.

23andME: Among 23andMe research participants with a reported RLS phenotype, about 45% had a relative with at least 700 cM shared identical by descent (corresponding to first cousins, i.e., 3rd degree relatives). The final set of unrelated participants selected for the GWAS excluded 30% of the original unfiltered set.

We have added the following sentence to the Online Methods, section “Study populations and phenotype definitions”:

“The reported sample numbers are the final sample numbers after quality control.”

3. I found Supplementary Table 2 (“Heritability estimates of pooled and sex-specific GWAS and meta-analyses”) difficult to follow. Perhaps the addition of a table footnote explaining the different columns/methods would be helpful, as well as separating pooled from sex-specific into two tables.

AUTHOR RESPONSE: We thank you for this valuable suggestion. We restructured Supplementary Table 2, splitting it into Supplementary Table 2a reporting results for the pooled dataset and Supplementary Table 2b reporting results of the sex-specific analyses.

We also moved the results of the LDSC heritability analysis, which is referenced in the main text, to the first columns (B and C) in the excel file. The heritability estimates produced by other models are now further to the right in the table. We added the following footnote explaining the column names and providing links to further information about the software and heritability models:

h2	Heritability estimate
h2_se	Standard error of heritability estimate
L0	Log likelihood of null model
L1	Log likelihood of alternative model
AIC	Akaike Information Criterion for alternative model
AIC_NULL	Akaike Information Criterion for difference between null model vs alternative model
logSS	Summary statistics-based log likelihood ratio between null model and alternative model
df	Degrees of freedom
(L1-L0)/df	Likelihood ratio tests statistic

Prop_LDSC	Proportion of explained heritability by LDSC assumption in the LDSC-LDAK hybrid model
Prop_LDSC_se	Standard error of Prop_LDSC

models used are described in the Online Methods of the manuscript; further information can be found in the original papers describing LDSC and SumHer/LDAK:

<https://doi.org/10.1038/s41588-018-0279-5>

<https://doi.org/10.1038/ng.3211>

<https://doi.org/10.1038/s41588-020-0600->

y

and on the respective webpages of the tools: <http://dougspeed.com/snp-heritability/>
<https://github.com/bulik/ldsc/wiki>

4. In the prediction model, the results indicate a PRS x age interaction, with the PRS being more impactful at a younger age. Does this perhaps warrant an age-stratified GWAS?

AUTHOR RESPONSE: We agree that conducting an age-stratified GWAS would undoubtedly yield valuable information. However, we think this needs to be done in a separate study with a specific focus on age-dependent genetic associations. Such an endeavor necessitates even more, especially younger individuals to be recruited, genotyped, and analyzed. At present, we do not have this type of dataset readily available. Moreover, we feel that such an investigation would extend beyond the scope of our present work, which

is already quite comprehensive. Even if we were to conduct an age-stratified GWAS with suboptimal data now, it would not be possible to present and discuss the results appropriately in the current manuscript.

Therefore, we have decided not to perform an age-stratified GWAS now, but we plan to conduct in-depth studies of age effects in RLS in the future.

5. For the sex-specific effects, there is an increased heritability in women. The authors explain this as an unobserved environmental risk factor. What would be the next steps to identify what this risk factor might be?

AUTHOR RESPONSE: Epidemiological studies have consistently demonstrated a clear sex difference in the prevalence of RLS, with women being affected twice as often as men. Two studies have suggested parity as a driving force of this difference

([doi:10.1001/archinte.164.2.196](https://doi.org/10.1001/archinte.164.2.196), <https://doi.org/10.1016/j.sleep.2009.04.005>). Renal disease and iron-deficiency are two further conditions with an increased RLS prevalence, which are also more frequent in women compared to men (<https://doi.org/10.1002/ajh.23397>, <https://doi.org/10.5664/jcsm.2664>, <https://doi.org/10.1111/ejh.12776>, <https://doi.org/10.1038/nrneph.2017.181>). We consider these “inner” environmental risk factors as interesting candidate risk factors to be prioritized in future studies.

To follow up on our model of GxE interaction, cohorts with detailed phenotyping are needed. This type of cohort is not available at present for RLS. RLS cohorts with detailed phenotypes are small or lack information on potential risk factors such as parity, while large population studies such as UK biobank still lack a detailed high-quality RLS phenotype. Therefore, the RLS research community needs to ask in a concerted manner for integration of precise RLS phenotyping in the ongoing large population studies.

We have revised the Discussion to stress the need for cohorts with extensive health and lifestyle data to study the role of age or other environmental factors and their interactions in RLS (please see the revised manuscript files).

6. With tissue and pathway enrichment demonstrating developmental pathways, is there any thought to why RLS is a later onset disease? Might there be earlier subclinical impacts? Is early (e.g., infant) genetic screening warranted?

AUTHOR RESPONSE: We thank you for this insightful comment. It’s important to note that RLS can manifest at any age, including childhood (<https://doi.org/10.1016/j.jsmc.2023.01.008>). However, the prevalence of RLS does tend to increase with age. The reasons for this observation are still being investigated, but there are a few potential explanations. Normal aging impacts on the functionality of all physiological processes in the human body, also affecting the central nervous system (CNS) (<https://doi.org/10.1016/j.cmet.2018.05.011>; <https://doi.org/10.1016/j.cell.2022.11.001>).

While no obvious signs of neurodegeneration have been found in RLS patients, age-related changes in dopamine function (alterations in dopamine signaling or changes in the sensitivity of dopamine receptors) could contribute to the onset of RLS. Environmental factors, e.g., diseases such as iron deficiency or medication use, may also contribute to disease onset.

These influences may become more prevalent with age, and their presence can contribute to the development or worsening of RLS symptoms. Additionally, certain medications

commonly used by older individuals, such as antihistamines or antidepressants, can also trigger or exacerbate RLS symptoms.

We hypothesize that the identified genetic risk variants lead to subtle changes in the CNS during early development thereby creating a susceptibility background in an individual. Aggravating factors, as well as gene-environment interactions, then accumulate during the lifetime until a threshold is reached and the CNS cannot compensate anymore (see also review on new concepts of secondary RLS: <https://doi.org/10.1212/WNL.0000000000002542>). However, to date there is insufficient data to verify this hypothesis. We are lacking longitudinal studies collecting rich information on lifestyle as well as RLS phenotype.

The same is true for genetic screening. While our case-control study indicates a small subset of individuals with an RLS odds ratio that comes close to the odds ratios observed in monogenic disorders for which infant screening is performed, the predicted risk needs to be verified in longitudinal studies. Moreover, it needs to be verified that preclinical intervention would ameliorate the course of RLS before preclinical screening is warranted.

7. For the causal relationships, the bidirectional relationship between RLS/insomnia/depression is mentioned as a complicated relationship which may include undiagnosed RLS cases in these other phenotypes. Do the authors think GWAS by subtraction (eg. GenomicSEM) might reveal unique genetic components? Or are there other methods to distinguish the undiagnosed RLS cases?

AUTHOR RESPONSE: We agree that the shared and unique genetic components of RLS and insomnia as well as RLS and depression, need to be discerned. However, the suggested method will require careful design in the case of RLS. While GWAS-by-subtraction with GenomicSEM will indicate shared as well as specific genetic components of RLS and e.g. insomnia, since the underlying model assumes the existence of these three components, it does not take into account heterogeneity due to misclassification, e.g. RLS cases being diagnosed as insomnia cases, creating a “mixed” component of RLS+insomnia. A successful application of this method helped to discern the roles of cognitive and non- cognitive genetics in educational attainment (<https://doi.org/10.1038/s41588-020-00754-2>). That analysis was based on a very clear model of $Edu \sim Cog + nonCog$. In the case of RLS, however, an analogous decomposition in insomnia and non-insomnia effects would be error- prone due to the high likelihood of misclassification.

We intend to study the genetic overlap between RLS and especially insomnia in future projects. Previously, we and others have successfully used BUHMBOX

(<https://doi.org/10.1038/ng.3572>) to detect evidence for a hidden subgroup due to misclassification of RLS in insomnia (<https://doi.org/10.1038/ng.3888>, <https://doi.org/10.1038/s41588-018-0333-3>), enabled mainly by the strong effect seen for the lead SNPs in *MEIS1* in RLS. An extension of this analysis using the larger dataset may provide additional insights. However, such statistical approaches are limited by the data available for the input studies. Ideally, samples would have clinical-grade or validated-questionnaire-based phenotype information for all diseases of interest. Currently, this is not the case. Our RLS cases are not systematically phenotyped for depression or insomnia symptoms and GWAS of insomnia symptoms have limited information on RLS phenotype in their cohorts. Improving the phenotyping of our RLS cases as well as increasing awareness

for including RLS in the phenotyping pipeline of biobank cohorts are among our current efforts to create better datasets. Hopefully, the sleep questionnaire which was started last October in UK biobank will also provide additional data to address this question.

For the current study under review, however, we feel that addressing the question of shared and non-shared genetics between RLS and insomnia is out of scope. As with your comment #4 about age-stratified GWAS, we do not believe that there is sufficient space in the current manuscript to present and discuss results of such an analysis appropriately. We think the overlap and differences in the genetic basis of RLS and insomnia warrants a study of its own rather than being presented as a sideline result in our current study.

8. Fig 1 is a little confusing and hard to read the underlying pathway names.

AUTHOR RESPONSE: We have adapted Figure 1 by resizing the pathway names and shifting the location of the respective eponymous pathway (highlighted in bold white font) for each pathway cluster. Moreover, we have edited the legend for more clarity.

Adapted Figure 1:

Adapted Figure 1 legend: “Treemaps of significantly enriched (FDR < 0.05) pathways. Respective GO terms were clustered based on their semantic similarity (method: Wang, GoSemSim as implemented in rrvgo package) using a) results from DEPICT and b) results from MAGMA. Terms are presented in rectangles. Coloring indicates the membership of a term in a specific cluster. In addition, each cluster is visualized by thick border lines. The size of each rectangle corresponds to the significance of the enrichment. The most significantly enriched term in each cluster was selected as the representative term and is displayed in bold white font.”

9. The colors used in Fig 5 make it hard to distinguish the lines.

AUTHOR RESPONSE: We have adapted Figure 5 by using the ColorBrewer Set 2 palette in R for better distinction of the lines. The ColorBrewer Set 2 palette consists of eight colors that are designed to be easily distinguishable and suitable for representing categorical data. The colors in Set 2 are chosen to have good contrast and to be perceptually distinguishable for most people.

Adapted Figure 5:

10. The grayscale in Supp Fig 5 makes it hard to differentiate the three bars.

AUTHOR RESPONSE: We have adapted Supp Figure 5 by enlarging the spacing of the bars and changing the colors.

Adapted Supp. Figure 5:

Reviewer #2:

Remarks to the Author:

Schormair et al present a genome-wide association study meta-analysis of 3 studies (EU-RLS-GENE consortium, INTERVAL study, 23andMe) for Restless Legs Syndrome (RLS), comprising of 116,403 cases and 1,473,166 controls of European ancestry. The authors report identifying 161 loci on the autosomes, with 131 novel and all 22 loci previously reported confirmed at $P < 5 \times 10^{-8}$. After conditional analysis, a total of 193 independent SNPs across the 161 loci were identified. In addition, sex-stratified analysis was also

performed.

The authors subsequently perform gene-set enrichment analysis to identify pathways implicated in RLS based on the GWAS summary stats. In addition, the authors use fine-mapping, genomic annotations, and co-localization analyses to prioritise candidate causal genes. Analysis of genetic correlations were performed to prioritise traits for bi-directional MR analysis prior. Finally, the authors present analyses of the development of a genetic predictor using a linear models (with and without interaction terms), random forests and deep neural networks.

Overall, this does provide some advances in the understanding of the genetic architecture and pathways involved in RLS. However, I do have some comments for the authors:

Introduction:

“However, a substantial proportion of heritability in RLS remains 15 unaccounted for. The implications of this are that polygenic risk scores (PRS) for RLS underperform...”

1. *OK but numerous studies have now shown this is unlikely to be fully resolved without inclusion of individuals of diverse ancestries – not Europeans. This is only mentioned right at the very end.*

AUTHOR RESPONSE: We fully agree that genetic studies of RLS should be extended to non-European populations and that this is an important limitation of our study as well as of genetic research in general. However, we could not extend our analyses beyond European populations in our present study due to the current lack of large-scale GWAS for RLS in populations of non-European ancestry.

Considering your comment and the related comment #1 of reviewer 1, we have restructured the introduction and discussion. We tried to focus the introduction on topics which we could address in our current study with the data available to us, while addressing study-specific limitations as well as general limitations of RLS research in the discussion. We do not intend to downplay the importance of including diverse ancestries, but we feel that the discussion section is the more suitable location to bring these limitations to the reader's attention.

In line with this view, we have adapted Introduction and Discussion of the manuscript accordingly (please see the revised manuscript files).

2. *“close to 1.5 million*

*controls” Give actual number
(1,473,166).*

AUTHOR RESPONSE: We now report the actual number in the abstract. While revising the manuscript, we noticed that the sample size numbers were partly incorrect due to an oversight on our side while dealing with different versions of the manuscript. We have now checked and corrected the numbers throughout the manuscript.

3. Can the authors expand on what they mean by “In order to provide opportunities for immediate translation into clinical use”? There have been few examples of this from GWAS. For example, a potential drug target is no sure guarantee of a compound being developed that passes all phases of a clinical trial. I also do not see how performing MR would lead into immediate translation either.

AUTHOR RESPONSE: We thank you for this insightful comment regarding the statement "In order to provide opportunities for immediate translation into clinical use" in our manuscript.

We appreciate the opportunity to clarify our intentions and acknowledge the points you raised.

By mentioning the goal of immediate translation into clinical use, we aimed to emphasize the potential practical applications of our research findings. While we understand that few examples exist of direct translation from GWAS to clinical practice, we believe it is important to strive for research outcomes that can eventually benefit patients. Although the development of a therapeutic intervention based on GWAS findings is a complex and multifaceted process, uncovering potential drug targets or biological mechanisms through genetic studies can lay the foundation for future drug discovery and development efforts.

We acknowledge that the path from identifying a potential drug target to the successful development of a clinical therapy is challenging and uncertain. Numerous factors, including compound optimization, preclinical testing, rigorous clinical trials, and regulatory approval processes, must be considered and navigated. We apologize if our statement may have inadvertently created the impression that immediate translation into clinical practice is a straightforward or guaranteed outcome.

Regarding your comment on performing Mendelian randomization (MR) and its potential for immediate translation, we agree that MR is primarily a method used to assess causality between traits or identify potential mediators in a causal pathway. While MR itself does not lead to direct translation, it can provide valuable insights into the underlying biology and

inform future experimental studies or clinical investigations. By identifying causal relationships or highlighting specific pathways influenced by genetic variants, MR can guide further research and help prioritize targets for therapeutic intervention.

We appreciate your perspective on the challenges of translating genetic research into clinical applications and the complexities associated with drug development. It is important to maintain a realistic outlook while also striving for scientific advancements that have the potential to impact patient care positively.

Based on your comments, we have revised the statement in question to ensure clarity and to align it with the limitations and complexities involved in the translation of GWAS findings to clinical practice as follows:

“To provide entry points for translational research, we prioritized drug targets among candidate genes, applied machine learning to optimize RLS risk prediction, and performed large-scale genetic correlation and Mendelian Randomization (MR) analyses to identify causal risk factors.”

Results

- Pooled autosomal GWAS analysis:

4. Can the authors hypothesise why the genetic correlations were different among the 3 studies – for example, is this likely due to the heterogeneity in the phenotype definition, differences in analytical software/models applied, etc?

AUTHOR RESPONSE: Thank you for raising an important point which we may not have addressed sufficiently in the manuscript. We appreciate the opportunity to provide more insight on the three studies included in our meta-analysis.

The three studies differ from each other in two phenotype-related aspects. Each study used a different phenotyping method to classify individuals as RLS cases, ranging from an online question to a face-to-face clinical interview. This is also due to the respective type of population sampled in each of the studies. 23andMe research participants are drawn from the general population, INTERVAL is a study conducted in healthy blood donors, and the EU-RLS-GENE study recruited its cases in specialized clinics for movement and sleep disorders.

All three GWAS included only European-ancestry individuals. Therefore, population

background is unlikely to be an important source of heterogeneity. LDSC with the EUR LD reference panel has been shown to be valid for genetic correlation and heritability analyses across European subpopulations, European-American, and UK populations as it does not introduce bias in the estimates (Bulik-Sullivan et al. Nature Genetics 2015, <https://doi.org/10.1038/ng.3211>).

Following your question about a potential impact of the different analytical models used in the three studies, we now compared the three different GWAS models (SNPTEST, PLINK, SAIGE) in the EU-RLS-GENE dataset and checked the genetic correlation between their results. The genetic correlation between SNPTEST and PLINK results was 0.989 ± 0.006 , between SNPTEST and SAIGE results 0.998 ± 0.0130 , and between PLINK and SAIGE results 0.985 ± 0.0149 . As the coefficient of variation is less than 1.51%, the effect of the GWAS models on the genetic correlation calculation was negligible.

Therefore, we conclude that differences in phenotype definition as well as target populations for recruitment are the main cause of the observed differences in genetic correlation

between the studies. Details on the different phenotyping methods and study populations of the three studies are provided in the Online Methods section.

We have now added a statement about potential contributors to the heterogeneity in the Results section (subsection Genetic discoveries):

“This was most likely due to differences between the studies in phenotyping of RLS as well as in source populations targeted for recruitment. (Online Methods).”

Furthermore, we added the following sentence in the Discussion section:

“They also reflect the breadth of target populations for recruitment into GWAS, including clinical cohorts as well as samples from the general population.”

5. *What is meant by $(\lambda)_{(subscript "1000")}$?*

AUTHOR RESPONSE: $(\lambda)_{(subscript "1000")}$ refers to the genomic inflation factor λ , which was used as a standard method to assess and correct for potential population stratification in GWAS before the introduction of LDSC (<https://doi.org/10.1111/j.0006-341x.1999.00997.x>). Since λ scales with sample size, λ_{1000} was introduced to have a measure comparable across different studies (<https://doi.org/10.1038/ng1333>). It is the inflation factor for an equivalent study of 1000 cases and 1000 controls.

However, since both $(\lambda)_{1000}$ and the LDSC intercept assess the same quality criterion, population stratification, we do not have to report both. To avoid confusion for readers and for the sake of brevity and clarity of the manuscript, we have removed the $(\lambda)_{1000}$ and keep only the LDSC intercept.

The sentence now reads as follows: “An LDSC intercept of 1.072 (se=0.013) indicated that population stratification was negligible, and that the inflation of the test statistics was driven by the polygenic architecture of RLS.”

6. Have the authors tested the whether the differences in the heritability estimates between the three studies are statistically significant? For example, for the estimates given in the main text, I note that some of point estimates lie within 95%CI of some of the other estimates.

AUTHOR RESPONSE: Thank you for this useful suggestion. Indeed, some of the confidence intervals are overlapping. We have conducted a two-sample Z-Test based on the point estimates for the heritability comparisons. Heritability was largest in the EU-RLS sample. The difference was strongly significant in comparison to the 23andMe study ($p = 0.0012$) but only borderline significant in comparison to the INTERVAL study ($p = 0.073$) which may be due to the rather small size of the latter study.

We have adapted the corresponding sentence in the manuscript accordingly:

“Heritability (as assessed by LDSC) in the most stringently phenotyped study, EU-RLS-GENE, was higher (0.26, se=0.038) than in INTERVAL (0.17, se=0.051, $p_{EU-Interval}=0.073$, two-sample Z-Test) and in 23andMe (0.14, se=0.011, $p_{EU-23andME}=0.0012$).”

- *Sex-stratified autosomal GWAS and meta-analyses*

7. Please add “statistically” prior to the word “significant” (and throughout).

AUTHOR RESPONSE: Following your justified request to increase the precision in terminology, we needed to compromise between adding “statistically” to all occurrences of the word “significant” and staying within the allowed word count of Nature Genetics articles, without having to remove other relevant content. Therefore, we added “statistically” whenever there was no other accompanying term (e.g., genome-wide, nominally, p-value reported in brackets, FDR reported in brackets) linking the word “significant” to a statistical analysis. We also did not add the term in the discussion section since it is now made clear in the results section.

8. “but both other cohorts individually showed higher estimates for heritability in females

compared to males ($p = 0.07$ in EU-RLS-GENE; $p = 0.09$ in 23andMe, Supplementary Table 2).” - Linked to comment above about presentation of ST2, can the authors confirm which values in ST2 these P-values are based on, and can the authors confirm what test has been performed to determine heterogeneity of heritability estimates?

AUTHOR RESPONSE: From all models used in heritability estimation, the LDSC model showed the best fit to our dataset. Therefore, the main text refers to results of the LDSC analysis. We have adapted Supplementary Table 2 to make the focus on LDSC more evident (please also see our response to comment #3 of reviewer 1). As stated above for the heritability estimation in the pooled data, we use a two-sample Z-Test for the comparison of heritabilities. We have added this information to the manuscript as follows:

“The INTERVAL study sample size was too small for reliable application of LDSC, but both other cohorts individually showed higher estimates for heritability in females compared to males ($p=0.07$ in EU-RLS-GENE; $p=0.09$ in 23andMe, two-sample Z-Test, Supplementary Table 2a).”

9. *“Comparing the two sex-specific meta-analyses showed a high genetic correlation of 0.96 ($se = 0.018$), however, the remaining small divergence was significant ($p = 0.044$), Supplementary Table 2” – Can the authors confirm how this test was performed?*

AUTHOR RESPONSE: We used a one-sample Z-Test to test if there was significant difference to unity for the genetic correlation between the sex-specific meta-analyses. We have added the following information to the text:

“Comparing the two sex-specific meta-analyses showed a high genetic correlation of 0.96 ($se=0.018$), however, the remaining small divergence was significant ($p=0.044$, one-sample Z-Test, Supplementary Table 2b).”

10. *The last sentence starts “The most significant sex differences” – this is a bit counter intuitive if basing on heterogeneity of effect estimates rather than being statistically significant in one sex and not in another – for which there are three examples out of the six. Also, the authors should note that the difference for the loci highlighted are small (~ 0.05 units) and the effects are directionally consistent.*

AUTHOR RESPONSE: We agree that highlighting these two loci could result in readers giving too much attention to them while ignoring the other loci mentioned in the table. Therefore, we have removed this sentence from the text and report all significant loci in extended data table 1. This is also in line with keeping the manuscript concise and within

formatting requirements.

11. While I appreciate the effort that has gone into the simulation work, is it not possible that the reason for the differences in heritability estimates are more simply due to the proportion of males and females that come from each of the three studies, where each study has defined RLS using different measures with some having higher heritability – over and above some unknown potential GxE interaction? The authors should provide a breakdown of the N for each sex from each of the three studies as well.

AUTHOR RESPONSE: We appreciate your feedback and have made the requested updates. We have added the sample numbers for each sex from each study in the Online Methods section titled 'Study populations and phenotype description - discovery meta-analysis'. Additionally, we have included the numbers and proportions of females as well as female cases for each study in Supplementary Table 2c (see below). We apologize for an oversight in the initial manuscript, where we forgot to update sample sizes from an interim version of the manuscript to the final version. We now corrected these numbers.

Considering the potential confounding effect of different proportions of females and males across studies, we carefully examined the data. The proportion of females is quite similar across the different studies, and this similarity is also observed when considering the proportion of female RLS cases within each study. Therefore, we think that these cannot account for the observed substantial difference in heritability.

We have added the following Supplementary table 2c: Proportion of sexes per study.

	N total cases	N total controls	N female cases	N female controls	proportion of females	proportion of female cases
EU_pooled	7248	19802	4769	9380	0.52	0.66
23andME_pooled	105908	1502923	71364	824262	0.56	0.67
INTERVAL_pooled	3491	23741	2200	11230	0.49	0.63

- X-chromosome analysis

12. It is not clear what the importance of links to neuroticism and estrogen mediated transcription is and why this is in the results and not discussion. Can the authors

provide more narrative for links to RLS or place in discussion.

AUTHOR RESPONSE: In line with the comment to our description of the sex-specific loci (see our answer above), we agree that highlighting single loci is not providing essential information. In view of the word count limits of the manuscript, we deem it more important to address the topics highlighted in your comments 1 (limitation to European ancestry) and 4 (cause of observed heterogeneity) in the discussion. Therefore, we have removed this sentence from the text and only refer to the supplementary table.

- Assessment of lead variants in independent cohorts

13. “Assessment of lead variants in independent cohorts” These are not “independent cohorts” – they are out-of-sample and biases from each of the studies may carry through to these subsets. Please do not refer to these as “independent cohorts”.

AUTHOR RESPONSE: We appreciate your comment regarding the terminology used to describe the cohorts assessed in our study. Upon careful consideration, we agree that referring to these cohorts as 'independent' may not accurately capture the nature of their relationship to the discovery cohorts. We apologize for any confusion caused.

While two of the cohorts were indeed collected by the same provider as the discovery cohort (23andMe and INTERVAL), we acknowledge that they are not independent in the strictest sense. The third cohort, however, has been collected by completely independent researchers. Considering this, we agree that 'additional' more accurately describes the relationship between these cohorts and the discovery cohorts.

Therefore, we have made the necessary adjustments and replaced the term 'independent cohorts' with 'additional cohorts' throughout the manuscript. This modification better reflects the relationship between the cohorts and addresses the concern raised by the reviewer.

.

- Functional annotation and biological mechanisms

14. “Excluding 12 SNPs not reaching genome-wide significance in the joint analysis of 1 discovery and validation did not change the results. (Extended Data Figure 6).” – This does not seem to be the correct figure.

AUTHOR RESPONSE: Done. Thank you for bringing this to our attention. We apologize for the error in the figure reference and appreciate your careful review. We have made the necessary corrections in the revised manuscript.

The sentence regarding the exclusion of 12 SNPs not reaching genome-wide significance has been moved to its correct location in the Online Methods section under "Gene-set and pathway enrichment analyses, DEPICT." The updated sentence now reads as follows:

"Excluding 12 SNPs not reaching genome-wide significance in the joint analysis of discovery and validation did not change the main results (Supplementary Table 25)."

We apologize for any confusion caused by the initial placement of this sentence and appreciate your assistance in ensuring the accuracy of our manuscript.

- Genetic correlations and MR analysis

15. There are a number of places where subscripting should be clarified by the authors – for example, it is not clear what $a_{\text{RLS-diabetes2}} = 0.99$ refers to - is this supposed to be the causal effect estimate?

AUTHOR RESPONSE: We have adapted the text of the LHC-MR results in order to make the annotation clearer. In general, we replaced the notation $a_{\text{trait1-trait2}}$ to $a_{\text{trait1} \rightarrow \text{trait2}}$, using an arrow to indicate the direction of the effect. Moreover, we added a more detailed description of the notation to explain the different types of p-values and effect estimates given. The section now reads:

"For other traits, LHC-MR analysis could provide evidence for relationships being causal rather than due to confounding. In terms of unidirectional relationships, RLS showed a significant effect (defined as $p_{\text{FDR}} < 0.05$) on type 2 diabetes with an effect estimate of $a_{\text{RLS} \rightarrow \text{diabetes2}} = 0.99$ ($\text{se} = 0.06$, $p_{\text{FDR}} = 1.5 \times 10^{-68}$) and significant likelihood-ratio tests for effects being only causal ($p_{\text{LRT_causal_only}} = 8.5 \times 10^{-28}$) and effects only of RLS on type 2 diabetes ($p_{\text{LRT_only_RLS} \rightarrow \text{diabetes2}} = 2.9 \times 10^{-40}$). Unidirectional causal links to RLS with strong evidence were fresh fruit intake (negative effect on RLS risk with $a_{\text{fruit} \rightarrow \text{RLS}} = -0.33 \pm 0.08$, $p_{\text{FDR}} = 0.0002$, $p_{\text{LRT_causal_only}} = 2.2 \times 10^{-5}$, $p_{\text{LRT_only_fruit} \rightarrow \text{RLS}} = 2.3 \times 10^{-5}$) and being tense/highly strung as well as having had a headache in the last month (increasing effect on RLS risk

with $a_{\text{tense} \rightarrow \text{RLS}} = 0.44 \pm 0.06$, $p_{\text{FDR}} = 8 \times 10^{-12}$, $p_{\text{LRT_causal_only}} = 8.6 \times 10^{-9}$, $p_{\text{LRT_only_tense} \rightarrow \text{RLS}} = 4.2 \times 10^{-8}$ and

$a_{\text{headache} \rightarrow \text{RLS}} = 0.37 \pm 0.08$, $p_{\text{FDR}} = 2.9 \times 10^{-5}$, $p_{\text{LRT_causal_only}} = 1.2 \times 10^{-8}$, $p_{\text{LRT_only_headache} \rightarrow \text{RLS}} = 6.9 \times 10^{-7}$).

Bidirectional relations with evidence pointing towards only causal effects were found for five traits (all with $p_{\text{LRT_causal_only}} < 0.05$): Ease of getting up in the morning lowered RLS risk ($a_{\text{ease} \rightarrow \text{RLS}} = -0.3 \pm 0.06$, $p_{\text{FDR}} = 1.3 \times 10^{-6}$) and vice versa ($a_{\text{RLS} \rightarrow \text{ease}} = -0.09 \pm 0.02$, $p_{\text{FDR}} = 0.0002$).

The frequency of tenseness/restlessness in the last two weeks as well as two traits reflecting lung function (having ILD/COPD differential diagnosis) increased RLS risk and vice versa, with a stronger effect on RLS ($a_{\text{tenseness} \rightarrow \text{RLS}} = 0.62 \pm 0.07$, $p_{\text{FDR}} = 5.1 \times 10^{-16}$;

$a_{\text{RLS} \rightarrow \text{tenseness}} = 0.11 \pm 0.02$, $p_{\text{FDR}} = 3.3 \times 10^{-5}$; $a_{\text{COPDdiff} \rightarrow \text{RLS}} = 0.38 \pm 0.06$, $p_{\text{FDR}} = 2.2 \times 10^{-9}$;

$a_{\text{RLS} \rightarrow \text{COPDdiff}} = 0.12 \pm 0.03$, $p_{\text{FDR}} = 7 \times 10^{-5}$). For self-reported osteoarthritis, the effect from RLS was stronger: $a_{\text{osteoarthritis} \rightarrow \text{RLS}} = 0.46 \pm 0.19$, $p_{\text{FDR}} = 0.033$; $a_{\text{RLS} \rightarrow \text{osteoarthritis}} = 0.18 \pm 0.04$, $p_{\text{FDR}} = 1.9 \times 10^{-5}$.

⁵. We also performed IVW MR analyses with Steiger filtering and MR-Egger intercept assessment as a secondary analysis. The results were consistent for 14 traits, which included the unidirectional link between RLS and type 2 diabetes (**Fig. 4 and Supplementary Table 22**)."

- *Development and validation of a risk prediction model*

16. How do these models compare with widely used models for PRS generation (LDPRED2, PRSICE, etc.)?

AUTHOR RESPONSE: We agree that it is of interest to see how using genome-wide SNP data impacts prediction power compared to using selected SNPs only. Therefore, we have run LDpred2-auto with genome-wide SNP data in the EU-RLS-GENE dataset. We saw that prediction performance was reduced compared to using our set of 216 SNPs only: $AUC_{(\text{LDpred2-genome-wide})} = 0.66 \pm 0.019$ and $AUC_{(216 \text{ SNPs})} = 0.73 \pm 0.018$. This difference was statistically significant with $P = 0.0056$ (two-sample Z-Test).

We have added the results of the Ldpred2-auto analysis to figure 5 and Supplementary Table 24 and we have added the following statement in the results section of the manuscript:

“Genetic risk was calculated using individual SNP dosages of 216 selected genome-wide significant SNPs, because this score showed better performance than a score built on genome-wide data ($AUC_{(LDpred2-genome-wide)} = 0.66 \pm 0.019$, $AUC_{(216\ SNPs)} = 0.73 \pm 0.018$, $p=0.0056$, two-sample Z-Test).”

Other Comments:

17. Figure 1 needs to be much clearer – some text is impossible to read

AUTHOR RESPONSE: We apologize for any confusion caused by the initial version of the figure and appreciate your valuable input.

Based on your comment, we have made several improvements to enhance the clarity and readability of the figure. We have resized the pathway names to ensure better legibility and have adjusted the positioning of the eponymous pathway for each pathway cluster to improve visual alignment.

Furthermore, we have revised the legend to provide clearer guidance on how to interpret the figure. The updated legend now provides explicit information about the clustering method used, the representation of terms in rectangles, the coloring scheme for cluster membership, and the significance represented by the size of each rectangle. We have also emphasized the representative term for each cluster by displaying it in bold white font. We believe these changes significantly improve the legibility and comprehension of Figure 1.

Adapted Figure 1 legend: “Treemaps of significantly enriched (FDR < 0.05) pathways. Respective GO terms were clustered based on their semantic similarity (method: Wang, GoSemSim as implemented in rrvgo package) using a) results from DEPICT and b) results from MAGMA. Terms are presented in rectangles. Coloring indicates the membership of a term in a specific cluster. In addition, each cluster is visualized by thick border lines. The size of each rectangle corresponds to the significance of the enrichment. The most significantly enriched term in each cluster was selected as the representative term and is displayed in bold white font.”

Supplementary Tables

18. *Supplementary Table 2 should be re-ordered to make references from the main text easier to navigate when reading the table.*

AUTHOR RESPONSE: We restructured Supplementary Table 2. We split it into Supplementary Table 2a reporting results for the pooled dataset and Supplementary Table 2b reporting results of the sex-specific analyses. We also moved the results of the LDSC heritability analysis, which is referenced in the main text, to the first columns of the excel file

(B and C) and shifted the heritability estimates from other models to the right side of the table. Moreover, we added a footnote explaining the column headers.

19. *Supplementary Table 3:*
Some of the cells contain #REF!

AUTHOR RESPONSE: We checked the tables and corrected the errors resulting from typing an incorrect formula.

20. *A p-value of 6.833632e-482 is presented. Please would the authors check this*

AUTHOR RESPONSE: Done. This p-value is correct. The extreme p-values mentioned here and in the next comment result from the (for common variants) unusually strong effect of the *MEIS1* lead variant on RLS.

21. *Supplementary Table 4:*
P-values of 2.17332e-747 and 3.428426e-447 are present. Please would the authors check these.

AUTHOR RESPONSE: Done. These p-values are correct.

22. *Extended Data Figures*

The legend for Extended Data 3 starts "Extended data figure 2 legend"

AUTHOR RESPONSE: We have corrected this typo.

Decision Letter, first revision:

12th October 2023

Dear Barbara,

Your revised manuscript "GWAS meta-analyses of restless legs syndrome identify 164 risk loci, highlight sex-specific effects, and advance risk prediction and treatment" (NG-A62148R) has been seen by the original referees. As you will see from their comments below, they find that the paper has improved in revision, and therefore we will be happy in principle to publish it in Nature Genetics as an Article pending final revisions to satisfy Reviewer #2's remaining request and to comply with our editorial and formatting guidelines.

We are now performing detailed checks on your paper, and we will send you a checklist detailing our

editorial and formatting requirements soon. Please do not upload the final materials or make any revisions until you receive this additional information from us.

Thank you again for your interest in Nature Genetics. Please do not hesitate to contact me if you have any questions.

Sincerely,
Kyle

Kyle Vogan, PhD
Senior Editor
Nature Genetics
<https://orcid.org/0000-0001-9565-9665>

Reviewer #1 (Remarks to the Author):

I have read the revision submitted by Schormair et al and I am satisfied with the edits. I also appreciate the thorough response and discussion of future work directions provided by the authors. The article is improved by the revisions and I look forward to seeing it in print.

Reviewer #2 (Remarks to the Author):

I thank the authors for taking the time to thoroughly address my concerns.

I do have one issue remaining based on my previous comment about phenotypic heterogeneity across the studies. In addressing my comment, the authors have added the following to the results section:

“This was most likely due to differences between the studies in phenotyping of RLS as well as in source populations targeted for recruitment.”

While the authors have provided a reasonable case for this in the response to my comment, the authors should either 1) provide data to explain to the reader why this is the “most likely” reason, or 2) change the language to be less definite and state this was a possibility that was accounted for.

Author Rebuttal, first revision:

Response to reviewers

Reviewers' Comments:

Reviewer #1:

Remarks to the Author:

I have read the revision submitted by Schormair et al and I am satisfied with the edits. I also appreciate the thorough response and discussion of future work directions provided by the authors. The article is improved by the revisions and I look forward to seeing it in print.

AUTHOR RESPONSE: We would like to thank you for your diligent revision of our manuscript and the insightful comments which allowed us to improve our manuscript substantially.

Reviewer #2:

Remarks to the Author:

I thank the authors for taking the time to thoroughly address my concerns.

I do have one issue remaining based on my previous comment about phenotypic heterogeneity across the studies. In addressing my comment the authors have added the following to the results section:

“This was most likely due to differences between the studies in phenotyping of RLS as well as in source populations targeted for recruitment.”

While the authors have provided a reasonable case for this in the response to my comment, the authors should either 1) provide data to explain to the reader why this is the “most likely” reason, or 2) change the language to be less definite and state this was a possibility that was accounted for.

AUTHOR RESPONSE: We would like to thank you for your diligent revision of our manuscript and the insightful comments which allowed us to improve our manuscript substantially.

We have adjusted our statement accordingly in order to point out that we are discussing this only as a possibility, not a definitive answer to the question. Due to the constraints put on the maximum length of the manuscript main text and online methods, we decided to change the language to be less definite. The sentence now reads:

“Genetic correlations between the three GWAS were strong but indicated some degree of heterogeneity, with pairwise genetic correlation (r_g) ranging between 0.70 and 0.76 (Extended Data Fig. 2), possibly due to differences between the studies in phenotyping of RLS as well as in source populations targeted for recruitment.”

Final Decision Letter:

19th April 2024

Dear Barbara,

I am delighted to say that your manuscript "Genome-wide meta-analyses of restless legs syndrome yield insights into genetic architecture, disease biology, and risk prediction" has been accepted for publication in an upcoming issue of Nature Genetics.

Your paper will be published online after we receive your corrections and will appear in print in the next available issue. You can find out your date of online publication by contacting the Nature Press Office (press@nature.com) after sending your e-proof corrections.

You may wish to make your media relations office aware of your accepted publication, in case they consider it appropriate to organize some internal or external publicity. Once your paper has been scheduled, you will receive an email confirming the publication details. This is normally 3-4 working days in advance of publication. If you need additional notice of the date and time of publication, please let the production team know when you receive the proof of your article to ensure there is sufficient time to coordinate. Further information on our embargo policies can be found here: <https://www.nature.com/authors/policies/embargo.html>

Before your paper is published online, we will be distributing a press release to news organizations worldwide, which may very well include details of your work. We are happy for your institution or funding agency to prepare its own press release, but it must mention the embargo date and Nature Genetics. Our Press Office may contact you closer to the time of publication, but if you or your Press Office have any enquiries in the meantime, please contact press@nature.com.

Acceptance is conditional on the data in the manuscript not being published elsewhere, or announced

in the print or electronic media, until the embargo/publication date. These restrictions are not intended to deter you from presenting your data at academic meetings and conferences, but any enquiries from the media about papers not yet scheduled for publication should be referred to us.

Please note that Nature Genetics is a Transformative Journal (TJ). Authors may publish their research with us through the traditional subscription access route or make their paper immediately open access through payment of an article-processing charge (APC). Authors will not be required to make a final decision about access to their article until it has been accepted. Find out more about Transformative Journals

Authors may need to take specific actions to achieve compliance with funder and institutional open access mandates. If your research is supported by a funder that requires immediate open access (e.g. according to Plan S principles), then you should select the gold OA route, and we will direct you to the compliant route where possible. For authors selecting the subscription publication route, the journal's standard licensing terms will need to be accepted, including <https://www.nature.com/nature-portfolio/editorial-policies/self-archiving-and-license-to-publish>. Those licensing terms will supersede any other terms that the author or any third party may assert apply to any version of the manuscript.

If you have not already done so, we invite you to upload the step-by-step protocols used in this manuscript to the Protocols Exchange, part of our on-line web resource, natureprotocols.com. If you complete the upload by the time you receive your manuscript proofs, we can insert links in your article that lead directly to the protocol details. Your protocol will be made freely available upon publication of your paper. By participating in natureprotocols.com, you are enabling researchers to more readily reproduce or adapt the methodology you use. [Natureprotocols.com](http://natureprotocols.com) is fully searchable, providing your protocols and paper with increased utility and visibility. Please submit your protocol to

<https://protocolexchange.researchsquare.com/>. After entering your nature.com username and password you will need to enter your manuscript number (NG-A62148R1). Further information can be found at <https://www.nature.com/nature-portfolio/editorial-policies/reporting-standards#protocols>

Sincerely,
Kyle

Kyle Vogan, PhD
Senior Editor
Nature Genetics
<https://orcid.org/0000-0001-9565-9665>